# Switchable topological polar states in epitaxial BaTiO$_3$ nanoislands on silicon

Ibukun Olaniyan [1,2] ✉, Iurii Tikhonov[3], Valentin Väinö Hevelke[1,2], Sven Wiesner[1], Leifeng Zhang [4], Anna Razumnaya [5], Nikolay Cherkashin[4], Sylvie Schamm-Chardon [4], Igor Lukyanchuk[3], Dong-Jik Kim[1] & Catherine Dubourdieu [1,2] ✉

A fascinating aspect of nanoscale ferroelectric materials is the emergence of topological polar textures, which include various complex and stable polarization configurations. The manipulation of such topological textures through external stimuli like electric fields holds promise for advanced nanoelectronics applications. There are, however, several challenges to reach potential applications, among which reliably creating and controlling these textures at the nanoscale on silicon, and with lead-free compounds. We report the realization of epitaxial BaTiO$_3$ nanoislands on silicon, with a lateral size as small as 30-60 nm, and demonstrate stable center down-convergent polarization domains that can be reversibly switched by an electric field to center up-divergent domains. Piezoresponse force microscopy data reconstruction and phase field modeling give insight into the 3D patterns. The trapezoidal-shape nanoislands give rise to center down-convergent lateral swirling polarization component with respect to the nanoisland axis, which prevents the formation of bound charges on the side walls, therefore minimizing depolarization fields. The texture resembles a swirling vortex of liquid flowing into a narrowing funnel. Chirality emerges from the whirling polarization configurations. The ability to create and electrically manipulate chiral whirling polar textures in BaTiO$_3$ nanostructures grown monolithically on silicon holds promise for applications in future topological nanoelectronics.

The investigation of ferroelectrics at the nanoscale in the past 10 years has unveiled a wealth of different whirling polar textures[1,2], such as vortex[3,4], flux-closure[5,6], skyrmion[7], meron[8], vortex-antivortex pairs[9,10], polar waves[11,12], bubble[13,14] and center-type domains[15,16]. These polar topological domains have raised great interest not only due to the fascinating physics of topology but also due to their potential for applications in future nanoelectronics. Their nanoscale dimensions (e.g., < 10 nm for skyrmions) make them attractive for ultrahigh-density data storage

(>Tb/in$^2$), if one can find a way to write/erase (or switch) the polar pattern they contain. Moreover, polar textures and their walls are regions of emerging properties such as negative capacitance[17,18] or chirality[19,20]. Negative capacitance is of particular interest in designing low-power field effect transistors[21]. Chirality, if it can be manipulated by an external stimulus (electrical or optical), has numerous applications in electronics or photonics. The formation of these whirling polar domains with complex topologies results from the interplay between electrostatic, polar gradient, and

[1]Helmholtz-Zentrum Berlin für Materialien und Energie, Hahn-Meitner Platz 1, 14109 Berlin, Germany. [2]Freie Universität Berlin, Physical and Theoretical Chemistry, Arnimallee 22, 14195 Berlin, Germany. [3]Laboratory of Condensed Matter Physics, University of Picardie, 80039 Amiens, France. [4]CEMES-CNRS and Université de Toulouse, 29 rue Jeanne Marvig, 31055 Toulouse, France. [5]Condensed Matter Physics Department, Jožef Stefan Institute, Jamova Cesta 39, 1000 Ljubljana, Slovenia. ✉e-mail: israel.olaniyan@helmholtz-berlin.de; dubourdieu@helmholtz-berlin.de

elastic energies[7,13,15] and therefore confinement, frustration and strain are key knobs that drive their formation in low dimensional ferroelectrics.

Experimentally, superlattices and trilayers containing PbTiO$_3$ or PbZr$_x$Ti$_{1-x}$O$_3$ (PZT) have served as the exemplary system for exploring polar textures, which vary with the substrate (mostly scandates or SrTiO$_3$) and the interlayer thicknesses[3,7,8,11,13]. In (PbTiO$_3$/SrTiO$_3$)$_n$ superlattices, vortices were stabilized on DyScO$_3$ single crystals[3], polar waves on GdScO$_3$[11], and skyrmions on SrTiO$_3$[7]. Bubbles were observed in PZT/SrTiO$_3$/PZT trilayers grown on SrTiO$_3$[13]. Additionally, merons were observed in PbTiO$_3$ thin films grown on SmScO$_3$[8]. To a lesser extent, exotic polar patterns have been also observed in Pb-free heterostructures. Periodic vortex arrays have been evidenced in BiFeO$_3$ layers sandwiched between scandates[22], and center-type domains have been observed in (BiFeO$_3$/SrTiO$_3$) superlattices on LaAlO$_3$ substrate[23]. For (BaTiO$_3$/SrTiO$_3$) superlattices grown on SrTiO$_3$, rotational polarization patterns have been reported[24].

At the other end of the material spectrum, nanoparticles (0D materials) are considered ideal candidates to host topological polar textures. BaTiO$_3$ nanoparticles embedded in a conducting non-polar polymer, studied by Bragg coherent diffraction imaging, have been shown to exhibit a 3D vortex that can be manipulated under an electric field[25]. Recently, BaTiO$_3$ nanoparticles analyzed by atomic electron tomography using scanning transmission electron microscopy revealed a 3D polar topological ordering[26]. However, manipulating nanoparticles and using other characterization methods such as piezoresponse force microscopy (PFM) is extremely challenging, often requiring embedding the nanoparticles in a matrix to form a composite material[25,27].

Ferroelectric epitaxial nanostructures such as nanodots, nanodisks, or nanocylinders would be better suited than superlattices for integration into nanoelectronics devices. Theory and modeling have predicted complex multidomain structures[28] and vortices in nanocylinders[29], vortices, and skyrmions in PbTiO$_3$ nanodots[20,30], center-type domains in PZT nanodots[31], and skyrmion-like domains in PZT nanoislands[32]. BaTiO$_3$ nanostructures of sub-10 nm in diameter have been predicted to host polar textures[33–38]. However, fabricating such nanostructures is extremely challenging[39]. Considering a top-down approach, etching or milling of complex oxides faces issues such as potential oxygen loss, compositional change, surface amorphization, and defects introduction[40–43]. This is particularly critical for etching structures of sub-100 nm size. The bottom-up approach, while offering the potential for creating smaller nanostructures with fewer defects, faces significant challenges in terms of achieving precise control over the size, shape, and uniformity of the nanostructures[39]. Additionally, the complexity of the growth processes and the difficulty in ensuring reproducibility have limited the widespread realization of this approach. BiFeO$_3$ nanoislands with sub-250 nm in lateral dimension have been achieved on SrTiO$_3$ substrate, revealing center-type polar domains[16]. Arrays of PZT nanodots deposited on SrTiO$_3$ substrate have been shown to exhibit vortex state through a comparison of PFM images and simulation results[4]. Diverse polar patterns such as flux-closure vortices, center-divergent vortices, center-convergent vortices, and antivortices have been imaged in PZT nanodots deposited on SrTiO$_3$ substrate by PFM[31].

Most of the exotic polar patterns reported in the literature have been realized on oxide substrates and with Pb-containing materials which are hazardous. Recently, skyrmion-like polar nanodomains have been obtained on silicon by growing an epitaxial trilayer on a SrTiO$_3$ substrate and transferring it as a membrane onto Si substrate[44]. Integration of polar texture on silicon, the ubiquitous electronic technology platform, remains challenging.

Here, we report the realization of BaTiO$_3$ chiral center-type down-convergent nanodomains on silicon. The polar topologies are hosted in nanoislands of truncated trapezoidal shape − like a narrowing funnel−embedded in an epitaxial BaTiO$_3$ film monolithically integrated into silicon. The domain patterns are studied by vertical and lateral PFM. Both PFM data reconstruction and phase-field simulations indicate a center down-convergent polarization confirmed by the Ti atom displacement map determined from scanning transmission electron microscopy (STEM). A swirling component around the nanoisland axis is evidenced, which confers chirality. The down-convergent nanodomains are reversibly switched to up-divergent ones by an external electric field. Phase-field modeling unveils the dynamical process of polarization switching with intermediate nonuniform swirled polarization states.

## Results and discussion

### Structural characterization of the epitaxial heterostructures and nanoislands

The different steps of the molecular beam epitaxy (MBE) leading to the occurrence of nanoislands in an epitaxial BaTiO$_3$ film are described in the Methods section. High-quality epitaxial BaTiO$_3$ films on Si using a SrTiO$_3$ template have been reported by several groups[45–50]. The first step is the direct epitaxy of SrTiO$_3$ on Si, which is achieved through the initial deposition of half a monolayer of Sr ((2 × 1) Sr surface reconstruction) aimed, in particular, at passivating the Si surface to avoid amorphous SiO$_x$ formation[51]. Here, we modified the passivation step of the Si surface by introducing an excess of Sr in order to disrupt the Sr (2 × 1) surface with Sr aggregates, which later trigger the nucleation of BaTiO$_3$ nanoislands embedded in the epitaxial BaTiO$_3$ film (Fig. 1a).

The 20 nm BaTiO$_3$/4 nm SrTiO$_3$ heterostructures on Si are single crystalline with the epitaxial relationship [110] BaTiO$_3$//[110] SrTiO$_3$//[100] Si and (001) BaTiO$_3$//(001) SrTiO$_3$//(001) Si (Figs. 1b, c and S1a of the Supporting Information). The average out-of-plane and in-plane lattice parameters of BaTiO$_3$ are 4.067Å and 3.976Å, respectively. Raman spectroscopy spectra (shown in Fig. S1b, Supporting Information) indicate that the BaTiO$_3$ film contains both c- and a-domains, with the long axis of the tetragonal cell perpendicular and parallel to the substrate plane, respectively. The B$_1$(TO) mode at 300 cm$^{-1}$ is a signature of the tetragonality of BaTiO$_3$[52]. The c-domains are evidenced by the E(LO$_3$) and E(LO$_4$) modes at 475 cm$^{-1}$ and 725 cm$^{-1}$, respectively, while the a-domains are evidenced by the A$_1$(TO$_3$) at 521 cm$^{-1}$[53]. The sharp peak of Si at 520 cm$^{-1}$ overlays the broad peak of the A$_1$(TO$_3$) mode.

The scanning electron microscopy (SEM) (Fig. 1d, e) and atomic force microscopy (AFM) (Fig. 1f, g) topography images reveal the presence of the BaTiO$_3$ nanoislands with a height of 3–6 nm and a diameter of 30–60 nm, whose density can be tuned by varying the Sr excess in the passivation step prior to SrTiO$_3$ growth (arrows are highlighting some of them in Fig. 1d, f). A high-resolution AFM image of two nanoislands is provided in Fig. S2 of the Supporting Information together with profile lines across them.

The BaTiO$_3$ nanoislands embedded in the film have a trapezoidal shape (Figs. 2a and S3 of the Supporting Information). The SiO$_x$ layer observed between Si and the SrTiO$_3$ layer (Fig. 2a, b) is formed after the epitaxial growth of SrTiO$_3$, during BaTiO$_3$ deposition because of oxygen diffusion down to the Si interface[49]. The high-angle annular dark-field (HAADF) high-resolution STEM image corrected from the distortions by the AbStrain method[54] (Fig. 2b) shows a closer view of a trapezoidal nanoisland. A bump-like feature is systematically observed below the nanoislands. These protrusions, that extend through the SrTiO$_3$ template, originate from the Sr excess introduced during the passivation step (prior to SrTiO$_3$ deposition) and lead to the nucleation of the BaTiO$_3$ nanoislands. The frontiers between each nanoisland and the film appear in dark contrast in the HAADF image (Fig. 2b). Similar dark-contrast regions in STEM images have been observed at the frontier between artificially assembled SrTiO$_3$ bicrystals with twist angles of 2–4°[55–57]. For a better visualization, Fig. 2c represents the

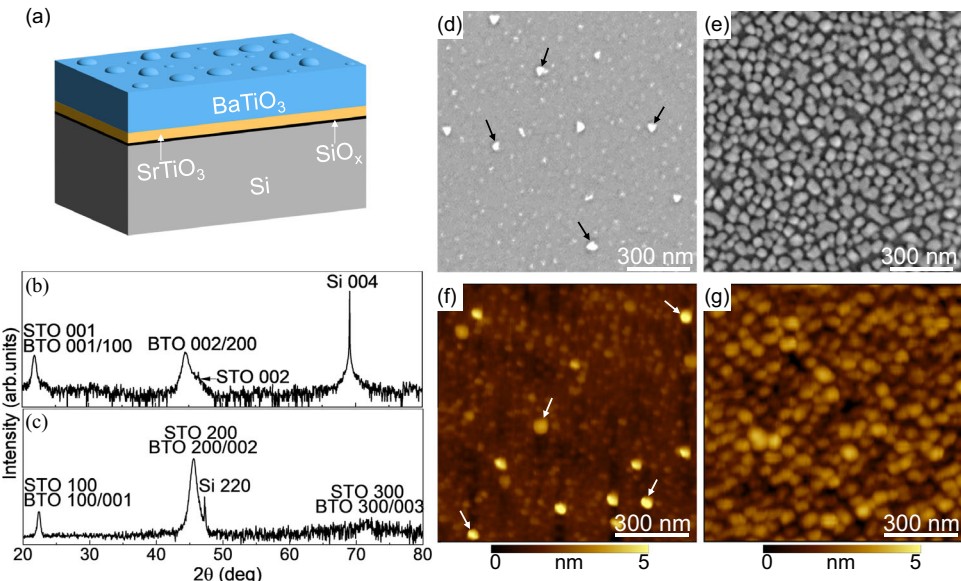

**Fig. 1 | Structural characterization of the sample. a** Schematic diagram of the epitaxial BaTiO$_3$/SrTiO$_3$/Si stack with nanoislands, (**b**) $\theta$/2$\theta$ X-ray diffraction pattern of the heterostructure, (**c**) in-plane X-ray diffraction pattern of the heterostructure, (**d**, **e**) SEM and (**f**, **g**) AFM images showing the two different densities of BaTiO$_3$ nanoislands for two different Sr fluxes in the passivation step. Some of the nanoislands in (**d**, **f**) are highlighted with arrows.

zoomed part of this region with the isolation of heavy atoms only (Sr and Ba), which was obtained by the Relative displacement approach[54]. These dark-contrast regions correspond to the growth planarity violation and appearance of tilt walls of edge dislocations with the [100] in-plane and the [001] out-of-plane Burgers vectors (Fig. 2c). Figure 2d shows the rigid-body rotation map obtained by AbStrain[54] from the HAADF high-resolution STEM image presented in Fig. 2b. As a result of the presence of walls of edge dislocations, larger than 1° rigid-body rotations of the crystalline structure are observed in the nanoislands and neighboring film. The rotation axis is perpendicular to the image plane, and the zero angle is defined as the normal direction of the SrTiO$_3$/Si interface. The rectangles in Fig. 2d indicate schematically the unit cell rotation. In the film, on the left side from the nanoisland, far enough from it, the BaTiO$_3$ unit cells are not rotated. When approaching the left wall of the nanoisland, the BaTiO$_3$ unit cells are clockwise rotated. In the nanoisland, a clockwise rotation increases in amplitude, reaching a maximum value of −2.2° close to the right wall. The BaTiO$_3$ unit cells in the film next to the right wall of the island are rotated counterclockwise. Such a difference in the rotations constitutes an abrupt interface between the island and the film around. The continuous rotation of the BaTiO$_3$ unit cell within the island allows to divide it into the left and the right half parts with opposite rotations with respect to its central axis. In the next step, we extract the relative displacement rotation map showing the direction of Ti atoms displacement from the barycenter position of the Ba(Sr) cells. For this, we have isolated an image of the heavy atoms (Ba and Sr) and an image of Ti atoms from the original HR-HAADF image presented in Fig. 2b and applied the Relative displacement procedure[54]. Figure 2e presents the resulting direction map, which relates to the polarization direction map. The white arrows indicate the principal values of the direction of the relative displacement vectors at different regions of the structure. One can see that such a map principally follows that of the rigid-body rotation. At the left and the right sides within the island, the relative displacement is mostly downward, with the inclination from the central axis of the island up to +20° counterclockwise and −15° clockwise at the bottom of the nanoisland.

Interface engineering with an excess of Sr in the Si surface passivation step prior to the SrTiO$_3$ buffer growth—proves to be an efficient way to induce nucleation seeds for the growth of epitaxial nanoislands embedded in a continuous epitaxial BaTiO$_3$ film. For a standard epitaxial SrTiO$_3$ film on Si (001), the reconstructed (2 × 1) Si surface, which consists of dimer rows, is passivated by ½ monolayer (½ ML) of Sr to avoid the formation of a SiO$_2$ amorphous oxide when oxygen is introduced in the MBE chamber for the SrTiO$_3$ growth, which would preclude any epitaxy on top[51]. The Sr flux is adjusted, following the RHEED pattern, to form a well-ordered Sr (2 × 1) surface. The Sr atoms sit in the trenches formed by the Si dimer rows, in the center of four dimers[58]. This is illustrated in Fig. S4a of the Supporting Information. We provide slightly more Sr than needed to form the ½ ML of Sr. It was shown by DFT calculations[59,60] that Sr atoms then occupy a position located in between two dimers, on top of a dimer row, as shown in Fig. S4b. In the study by Potočnik et al.[60], the passivation of the Si surface was performed by pulsed laser deposition, and non-intentional adatoms of Sr was evidenced by scanning tunneling microscopy imaging. Here, we provide additional Sr intentionally so that Sr atoms occupy these sites (in between two dimers, on top of a dimer row), and form protrusions on the Sr surface, which serve as nucleation centers to form the nanoislands. Our STEM study (see Fig. 2b) clearly shows that the nanoisland grows from a bump that is already present in the SrTiO$_3$ layer and originates underneath from the Si surface as a protrusion in the SiO$_x$ interfacial layer. The elemental profiles across such a bump are shown in Fig. S5 of the Supporting Information, as obtained from the acquisition of the corresponding EELS edges (O-K, Ti-L, Ba-M, Sr-L, and Si-K). Below the SrTiO$_3$ layer (in the bump), an additional Sr peak appears, corresponding to the excess Sr brought initially before the SrTiO$_3$ layer is grown. The bump is composed of a Ti-Sr-Si-O silicate resulting from the interdiffusion between the SiO$_2$, SrO (the excess Sr during the passivation phase is oxidized as soon as oxygen is sent) and the upper SrTiO$_3$ layer (diffusion of Ti).

The density of the nanoislands can be tuned by varying the Sr flux: the larger the Sr excess, the greater the density of Sr aggregates, which in turn leads to a higher density of nanoislands. Nanoislands may eventually merge, as shown in the SEM image of Fig. 1e. Self-assembled nanoislands have been reported in BiFeO$_3$ films grown on a LaAlO$_3$ substrate by interface modification[61]. The proposed approach here is attractive as the nanoislands do not appear as discontinued nanostructures but are part of a continuous matrix.

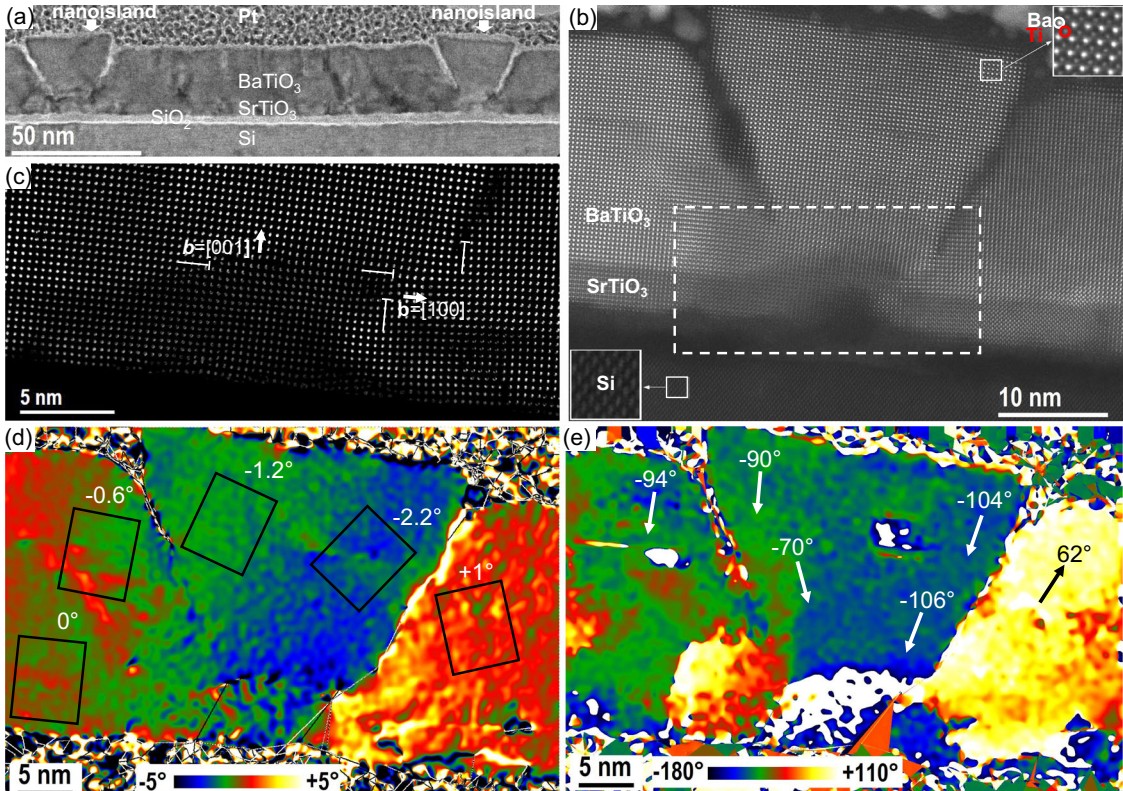

**Fig. 2 | STEM characterization of BaTiO₃ nanoisland. a** Cross-sectional (110)$^{Si}$ STEM image showing two nanoislands in the thin BaTiO₃ film. **b** Cross-sectional (110)$^{Si}$ high-angle annular dark-field high-resolution STEM image of the BaTiO₃ nanoisland seen on the right side of (**a**). The region marked in the dashed rectangular is a protrusion under the BaTiO₃ nanoisland, which originates from the excess of Sr and triggers the nucleation of the nanoisland. **c** The zoomed part of this region

is represented by heavy atoms only (Sr and Ba). Edge dislocations with in-plane and out-of-plane Burgers vectors are indicated. **d** Rigid-body rotation map showing the rotation of the unit cells (black rectangular) and (**e**) Relative displacement rotation map showing the direction of Ti atoms displacement from the barycenter position of the Ba(Sr) cells, in the nanoisland and the thin film. The STEM images were obtained from the sample with the lower density of nanoislands shown in Fig. 1d, f.

## Topological polar domains in the nanoislands

Figure 3 shows typical topographic, vertical-PFM (VPFM), and lateral PFM (LPFM) images recorded on a sample. VPFM provides information on the polarization component, which is along the vertical z-axis. LPFM detects only the in-plane component of polarization, which is perpendicular to the cantilever, via torsional vibration[62]. In the following, the vertical amplitude, vertical phase, lateral amplitude, and lateral phase are noted as V-amp, V-phase, L-amp, and L-phase, respectively. The nanoislands are clearly seen on the topographic image (Fig. 3e). At these locations, peculiar polar domains are observed (circled in white in the PFM images). The V-phase signal displays a uniform bright contrast (Fig. 3b), indicating that the vertical component of the polarization in the nanoisland points downwards ($P_{down}$), towards the BaTiO₃/SrTiO₃ interface. In the corresponding L-phase image (Fig. 3d), the same domains exhibit half-bright and half-dark contrast, with a phase difference of 180° (Fig. 3d), indicating two opposite components for the in-plane polarization. The boundary between these two regions of opposite contrast gives rise to the dark, thick line separating two bright polar regions in the L-amp image (coffee-bean-like pattern). Hence, the polarization pattern in the nanoislands has a vertical component and two opposite lateral components.

The bright contrast observed at the locations of the nanoislands in the V-amp images (Fig. 3a) can be ascribed to topographic crosstalk[63]. The question of topographic crosstalk is also raised, of course, in the lateral PFM images. To mitigate artifacts from cantilever torsion caused by the frictional force, we systematically set the fast scan direction along the long axis of the cantilever to minimize the cantilever torsion during the lateral PFM scans, as suggested by J. Kim et al.[64] We checked that the trace (forward) and retrace (backward) LPFM

images were identical (Fig. S6). Additional experiments such as rotating the sample (see Fig. 4) have been done to preclude major crosstalk effects. Note that a similar coffee-bean L-amp contrast has been reported in other systems, such as PbTiO₃/SrTiO₃/PbTiO₃ trilayers[44] and BiFeO₃ nanodots[15].

PFM imaging was performed after rotating the sample counterclockwise at various angles to gain information on the polarization along the three x, y, z directions of space. For the angle 0°, the cantilever is parallel to a <100> crystallographic direction of BaTiO₃/SrTiO₃, say [100] (parallel to [110] crystallographic direction of Si). At an angle of 90°, the cantilever is then parallel to the [010] direction (parallel to [1–10] crystallographic direction of Si). Throughout the rotation at angles of 0°, 30°, 60°, and 90°, the dark contrast in the L-amp images and the half-bright/half-dark contrast in the L-phase images remain parallel to the cantilever, regardless of the angle (Fig. 4). This observation indicates that the in-plane polarization has an approximate rotational symmetry. As expected, the vertical polarization component remains unchanged with rotation. To depict the polarization pattern, we assume that the bright and dark contrasts in the lateral phase images correspond to in-plane polarization pointing right and left, respectively. The opposite assumption (i.e., bright and dark contrasts corresponding to the polarization pointing left and right, respectively) does not impact our main conclusion. Two possible polar patterns possessing an in-plane rotational symmetry are considered: the flux-closure rotational polar texture, and the divergent/convergent polar texture (center-type), as illustrated in Fig. S7, Supporting Information. The flux-closure rotational polar texture (Fig. S7b) should have the in-plane domain wall perpendicular to the cantilever, while the center-type domain pattern (Fig. S7a) should have the in-plane domain

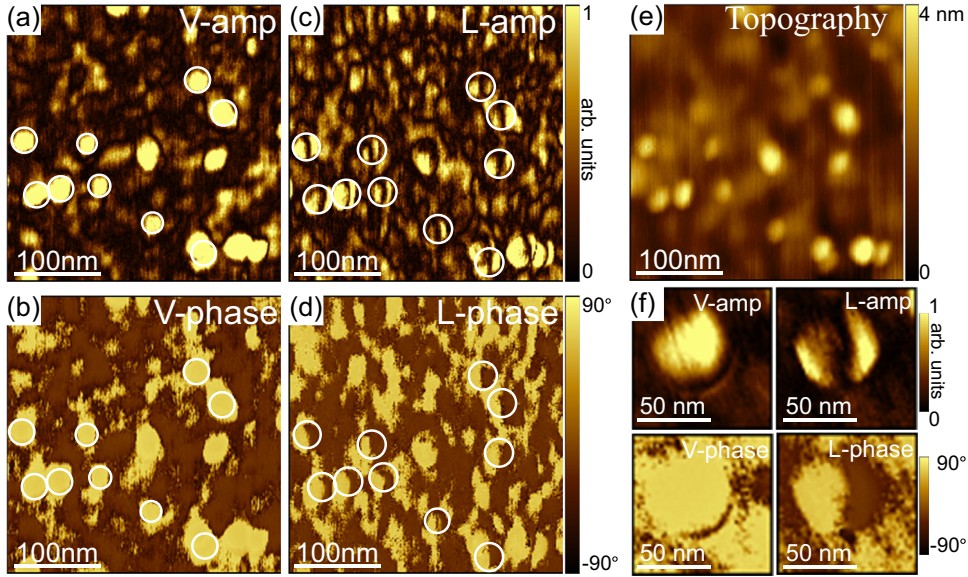

**Fig. 3 | Pristine PFM images of BaTiO₃ nanoislands. a** Vertical amplitude, (**b**) Vertical phase, (**c**) Lateral amplitude, and (**d**) Lateral phase contrast. **e** AFM topography image, (**f**) High-resolution PFM images of a nanoisland.

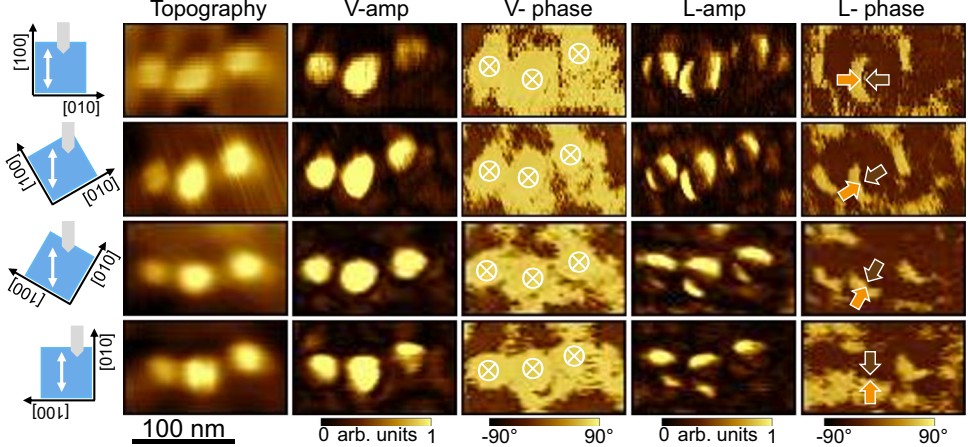

**Fig. 4 | PFM imaging of BaTiO₃ nanoislands at different in-plane angles.** Topography, vertical-PFM amplitude and phase, lateral PFM amplitude and phase images captured in the same region of a sample while rotating it at different in-plane angles. The sample orientation with respect to the cantilever is shown on the left side of the image. The white arrows represent the fast scan direction.

wall parallel to the cantilever, as indicated by the different colors in Fig. S7. Since the in-plane domain wall is always approximately parallel to the cantilever (Fig. 4), the polar texture in the nanoislands is center-type, with the vertical component of the polarization pointing down. At the same time, the slight deviation of the domain wall from a straight line, resulting in its coffee-bean structure, indicates a minor local buckling of polarization. The down-convergent polarization flux is consistent with the results obtained by STEM for the direction of Ti atom displacements, as shown in Fig. 2e.

Further insight into the 3D polarization texture in the nanoislands is given by phase-field modeling based on the PFM vector reconstruction, as illustrated in Fig. 5. The set of analyzed vertical, and 0°- and 90°-rotated lateral amplitude, and phase PFM images is shown in Fig. S8 of the Supporting Information. The PFM polarization vector reconstruction ($P_x$, $P_y$, and $P_z$)—for each spatial location X, Y—is a weighted average over the thickness of the nanoisland (with a large contribution from the surface and from underneath the surface).

The numerical vector reconstruction of the PFM data within the nanoisland area is presented in Fig. 5a–d (see Methods for details).

Figure 5a, b shows the top view of the reconstructed polarization field **P**, superimposed with color maps of the signs of the $P_x$ and $P_y$ components, respectively. These colorings correspond to the 0° and 90° PFM L-phase images shown in Fig. S8 of the Supporting Information. The reconstructed distribution of the polarization field demonstrates the central convergence of the in-plane polarization component, albeit with some buckling from the radial direction. Notably, this buckling corresponds to the helical clock- or counter-clockwise turning of the polarization vector around its average down-converging direction when moving through the thickness of the nanoisland. Respectively, this brings a local right-hand (positive) or left-hand (negative) chirality to the rotating patterns of the texture. The corresponding left- and right-screwed domains are discerned in Fig. 5c by the color map of the normalized chirality indicator, $\chi' = \boldsymbol{P}\bullet[\nabla\times\boldsymbol{P}]/<P^2>$[33]. Specifically, if $\chi$ is not equal to zero, it suggests that the volume configuration is chiral. However, the inverse is not necessarily true. Positive and negative volume chirality, if present, can cancel each other out by averaging throughout the nanoisland thickness, resulting in $\chi' = 0$. The side view of the

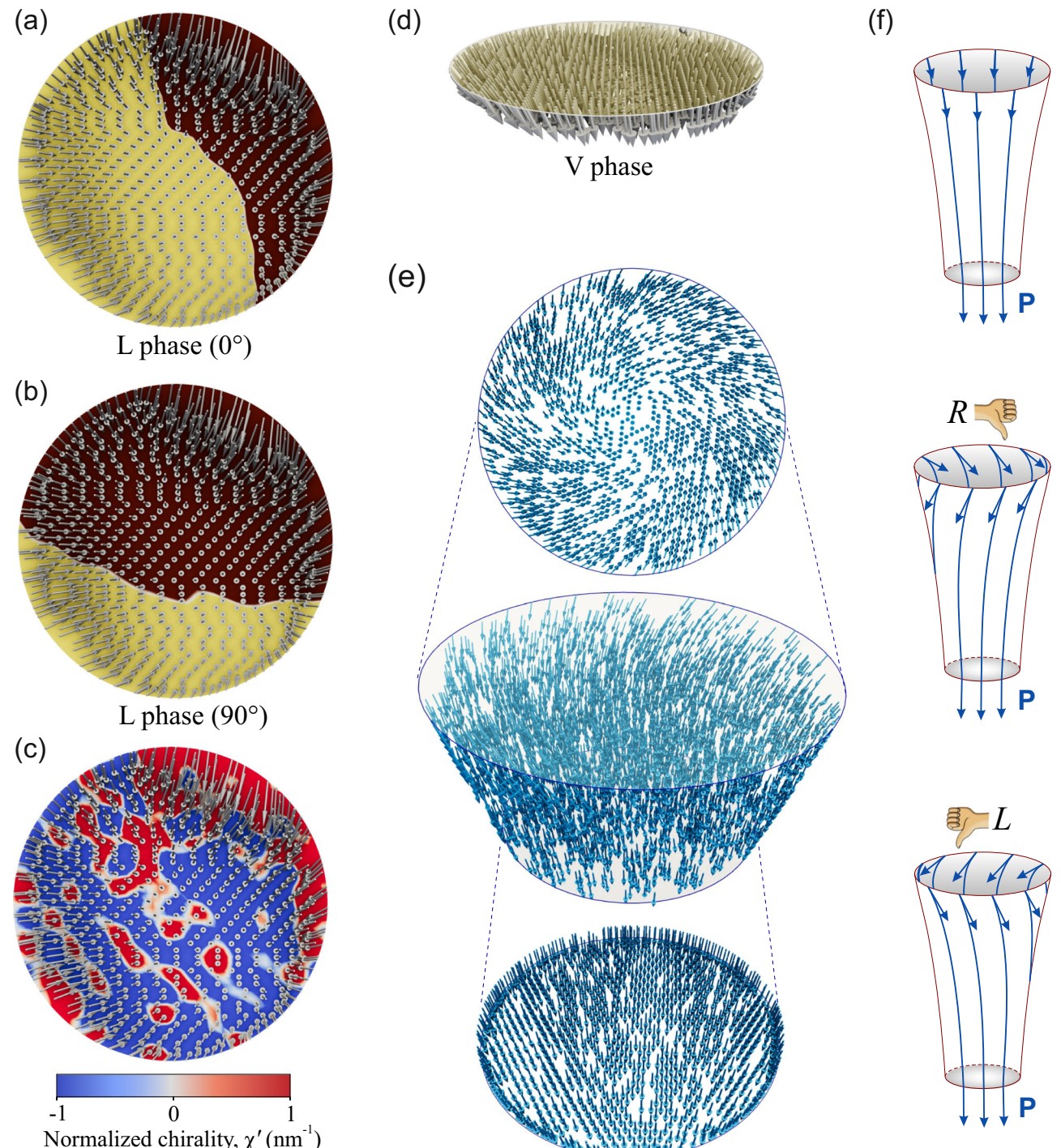

**Fig. 5 | Polarization structure in BaTiO₃ nanoisland. a** Top view of the reconstructed vector polarization field. The yellow and brown colors visualize positive and negative signs of $P_x$ component of polarization **P** corresponding to the 0° PFM L-phase image shown in Fig. S8 of the Supporting Information. **b** The same for $P_y$ component corresponding to the 90° PFM L-phase image, shown in Fig. S8 of the Supporting Information. **c** Top view of the reconstructed vector polarization field. The color map shows the normalized chirality indicator, $\chi'$. **d** Side view of the reconstructed polarization vector field in the nanoisland area. The yellow circle corresponds to the PFM V-phase image in Fig. S8 of the Supporting Information. **e** Modeling of the polarization texture in the nanoisland with top and bottom views. **f** Achiral (top), right-handed (middle), and left-handed (bottom) stream tubes of the polarization down-converging flux.

reconstructed polarization vector orientation, shown in Fig. 5d, confirms the central down-converging polarization texture.

To gain a deeper understanding of the experimentally observed pattern formation, we conducted a full phase-field simulation of the system based on the minimization of the Ginzburg-Landau energy, coupled with electric and elastic fields (see Methods section for details). The results shown in Fig. 5e demonstrate the center down-

convergent polarization field, resulting from the compression of the polarization flux by the down-contracting lateral sidewalls of the nanoisland. Importantly, the polarization vectors stay parallel to the sidewalls to prevent the formation of bound charges that would otherwise produce energy-unfavorable depolarization fields. This conclusion is corroborated by the STEM image presented in Fig. 2e, which shows Ti atom displacements parallel to the sidewall of the

nanoisland. Conductive AFM also allows us to rule out the presence of bound charges on the sidewalls, which would otherwise be associated with screening charges at the boundary between the nanoislands and the film. No enhanced conduction is observed in these boundaries, as shown in Fig. S9 of the Supporting Information. At the same time, polarization flux flows freely through the upper and lower surfaces of the nanoisland since the polarization-bound charges are screened by the conducting carriers of the bottom Si substrate and by the charges accumulated at the free top surface. An important observation from the phase modeling is the confirmation of the observed swirling component of the polarization around the nanoisland axis that was unveiled from the PFM data reconstruction. This component is more pronounced at the top of the nanoisland and vanishes at the bottom. The texture as a whole resembles a swirling vortex of liquid flowing into a narrowing funnel. The basis for this analogy[20,33] lies in the similarity of the flow of a liquid with the flux of polarization in ferroelectrics, resulting from their almost divergence-free character. While the absence of the divergence in the velocity field in liquids originates from their incompressibility, the divergence-free nature of the polarization field in ferroelectrics results from the minimization of the electrostatic depolarization energy produced by the volume-bound charges $\rho = -\nabla \cdot P$. Notably, in ferroelectrics, the divergence-free constraint is approximate, since part of the emerging volume-bound charge may be screened by the free carriers of charge.

Using the hydrodynamic analogy, we consider a stream tube formed by polarization streamlines inside a tapering nanoisland, as shown in Fig. 5f; the polarization flux is conserved throughout the entire length of the tube to prevent the formation of bound charges. When the streamlines are aligned with the tube axis, as in the top panel, the conservation of flux results in a decrease in polarization amplitude towards the top of the tube. Such configuration is not energetically favorable since polarization prefers to keep the constant magnitude of the spontaneous polarization. The more advantageous configurations, in which the polarization swirls at the wider part of the tube while keeping constant flux and magnitude, are shown in the middle and bottom panels. The swirling can occur in either a clockwise or counterclockwise direction, providing right-handed (R) or left-handed (L) chirality to the polarization texture inside the tube, respectively.

The simulated structure, shown in Fig. 5e, demonstrates unique left-handed chirality across the entire volume of the nanoisland. The emergence of domains with different chiralities observed in the reconstructed polarization pattern (Fig. 5c) is explained by the low energy barrier between these states. We conclude here that the chirality distribution inside the nanoisland may depend on the dynamic history of the polar phase preparation. Special care should be taken to achieve a monochiral state.

In addition to the center-type down nanodomains, another polar texture, more complex, has been occasionally observed, as shown in Fig. S10 of the Supporting Information. Note that the observation of multiple domain patterns in similarly shaped nanoislands is further evidence that the lateral PFM contrasts do not originate majorly from topographic effects[4]. Such non-central structures also occasionally emerge in the phase-field simulations. However, energy analysis shows that the energies of such textures are higher than those of the centered-type polarization formations obtained under thermal quenching or field cycling for the same sample geometries. We, therefore, conclude that these are dynamically formed metastable patterns in which the polarization texture stopped relaxing to a more advantageous central-type polarization state and became pinned at inhomogeneities.

The PFM study reveals a majority of center-type down-convergent nanodomains, which is in good agreement with phase-field modeling of the polarization pattern of truncated conical-shaped nanostructures. So far, center-type domains have been mainly found and explored in BiFeO$_3$ nanodots[9,15,16,65]. Center-type domains have also

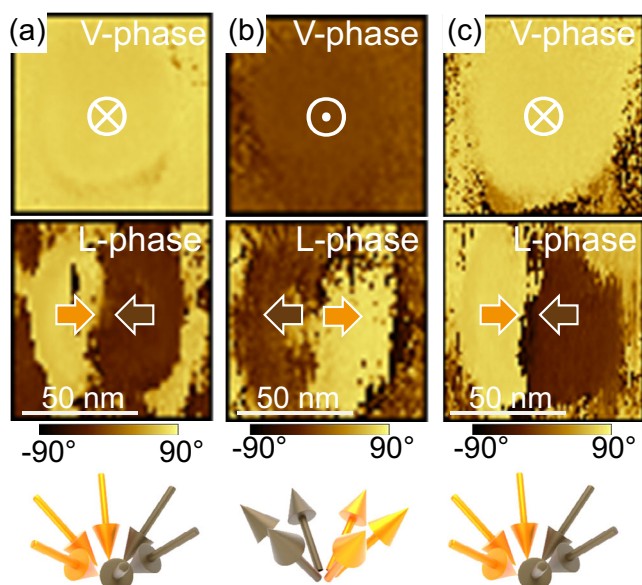

**Fig. 6 | Demonstration of polarization switching in BaTiO$_3$ nanoisland.** PFM vertical/lateral phase images and polarization schematics of (**a**) pristine down-convergent domain, (**b**) up-divergent domain after scanning with a biased tip (−9 V), and (**c**) down-convergent domain after subsequent scanning with a reversely biased tip (+9 V).

been induced by applying a radial electric field to BiFeO$_3$ films grown on DyScO$_3$ substrate[66], by oxygen vacancies in BiFeO$_3$ films grown on a SrTiO$_3$ substrate[67], and by increasing the mechanical load on PbTiO$_3$ films grown on SrTiO$_3$ substrate[68]. More recently, center-type domains were reported in a PbTiO$_3$/SrTiO$_3$ bilayer transferred onto Si substrate[44]. In addition to the center-type configuration, both the data reconstruction and the phase-field modeling show that the lateral polarization component is whirling. The nanoscale size of the islands, their shape, and their structural decoupling from the matrix, as well as the strain gradient, favor the occurrence of the whirling polar patterns to minimize the depolarization field.

## Electrical switching of the topological textures

The pristine PFM images of a nanoisland exhibiting a down-convergent center-type topological texture are shown in Fig. 6a, together with the sketch of the texture. After scanning the nanoisland with a tip bias of −9 V, the bright contrast of the V-phase image becomes dark, indicating that the vertical polarization component switches from down to up. The contrasts of the L-phase signal are also inverted, indicating the switching from a convergent to a divergent lateral polar configuration (Fig. 6b). Hence, the down-convergent texture can be switched to an up-divergent one as schematically shown in Fig. 6b. Equally exciting is that the up-divergent domain can be reversibly switched back to the down-convergent domain by scanning the nanoisland with a tip bias of +9 V. These topological polar nanodomains are reversibly switchable under an applied electric field.

A notable switching pathway in the domain pattern was observed in some cases during the electrical switching from a down-convergent to an up-divergent domain (Fig. 7a). Panel i of Fig. 7a shows the pristine state of a down-convergent domain. After one scan with the tip at −9 V (panel ii), the V-phase signal turns into a bubble-like core-shell pattern, with a dark contrast (P$_{up}$) core region surrounded by a bright contrast (P$_{down}$) shell region. The lateral phase component shows a core-center structure. After a second scan with a tip bias of −9 V bias (panel iii), the bubble-like domain pattern has fully switched to a monodomain P$_{up}$ and a divergent lateral component (images similar to those of Fig. 6b), namely the up-divergent domain. The intermediate state (panel ii)

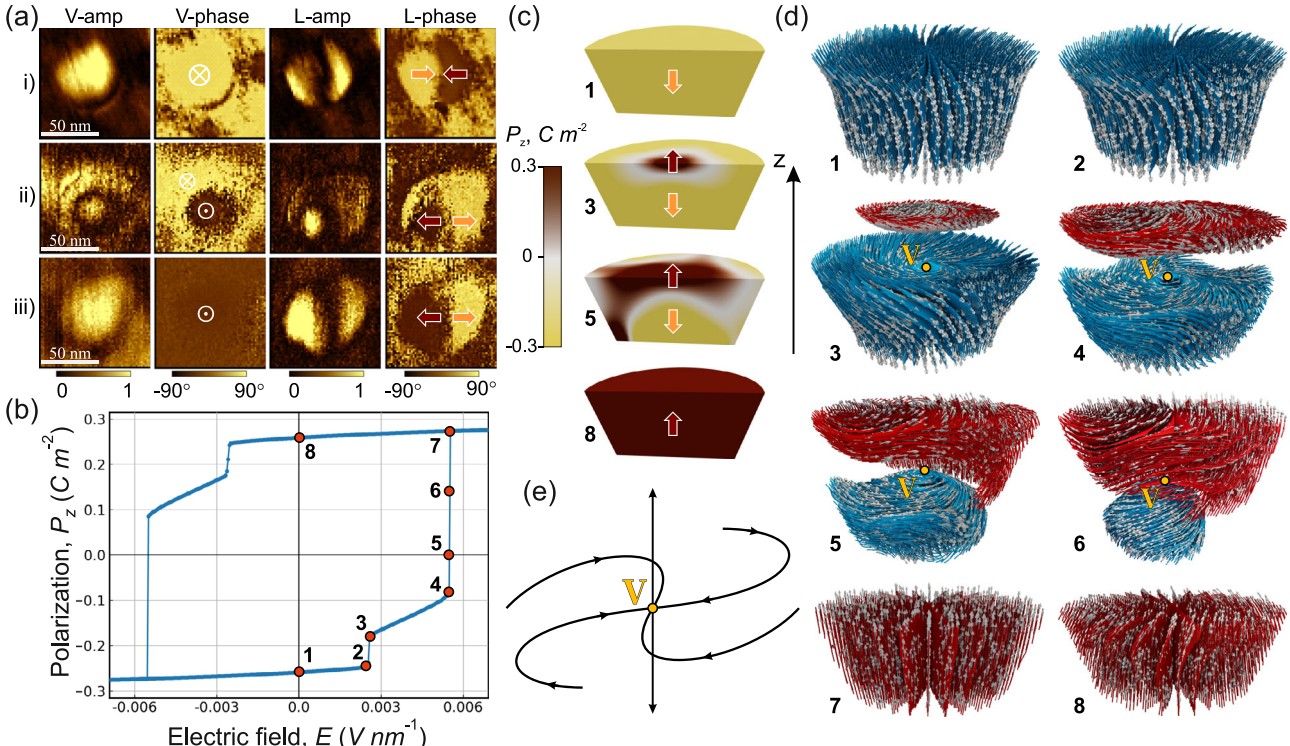

**Fig. 7 | Polarization switching in nanoisland. a** Experimental V-amp, V-phase, L-amp, and L-phase PFM images of the pristine state showing a down-convergent domain (panel i), after switching to a metastable state with bubble-like domain (panel ii) and after switching to an up-divergent domain (panel iii). Vertical and lateral PFM amplitudes are given in arbitrary units. **b** The *P-E* hysteresis loop of nanoisland switching, showing the simulation results. The numbers correspond to the states to be analyzed. **c** Distribution of the vertical polarization component, $P_z$,

in the initial (1), intermediate (3, 5), and final switched (8) states. The numbers correspond to the locations of the states on the *P-E* hysteresis loop. **d** Dynamics of the polarization vector field during the switching. The blue and red colors correspond to the polarization fluxes in unswitched and switched domains. The switched domain is slightly displaced upwards for the convenience of presentation. Point V is the singular point of the vector field where these fluxes meet. **e** Topological properties of the singular point V.

observed when switching from down-convergent to up-divergent textures bears resemblance to the ferroelectric bubble domains observed in PZT nanodots on a SrTiO₃ substrate[14], in a PZT/SrTiO₃/PZT trilayer grown on a SrTiO₃ substrate[13], and in chemically modified $Bi_{0.5}Na_{0.5}TiO_3$ bulk crystals[69]. In contrast to previous studies where bubble domains were observed in the pristine state, our bubble-like domains are formed during the electrical switching of down-convergent domains. Ferroelectric bubble domains have been theoretically predicted to occur as a transitional state between polarization vortex tubes and a monodomain in ultrathin PZT[70]. Here, the bubble-like domain is a transitional state between the two types of center-type domains, which is stable in time, as it could be imaged again after eight hours. It is important to note that the observed bubble-like domain is metastable. We do not consistently obtain it when switching from the down-convergent to the up-divergent domains or vice versa. This is expected as it has been reported that even slight external stimuli (such as tip-induced pressure, atmospheric conditions, etc.) are enough to destabilize the bubble domains[13].

**Dynamical process of polarization switching from phase-field simulations**

Now, we delve deeper into the dynamic process of polarization switching using phase-field modeling (see the Methods section for details). Figure 7b demonstrates the field-polarization hysteresis loop for the nanoisland obtained from simulations. Figure 7c shows the dynamics of reversing the direction of the vertical polarization component, and Fig. 7d gives details of the evolution of the whole polarization vector fields during the switching. The numbers in panels (c) and (d) correspond to the exemplary points on the hysteresis loop in panel (b).

The switching of the down-converging swirled polarization state (state 1) begins with further swirling of the polarization streamlines (state 2), followed by the nucleation of a round-shaped, oppositely oriented polarization domain (state 3–in red), located at the top of the nanoisland. This transition occurs due to the polarization jump from state 2 to state 3, as observed in the hysteresis loop in panel (b). The cap of the repolarized domain grows with increasing electric field, reaching a critical state (state 4), after which the polarization rapidly switches to an upward-diverging polarization state (state 7), dynamically passing through intermediate states 5 and 6. The swirling of polarization in state 7 is minor; however, when the field is removed, the system finally relaxes to the upward-diverging swirled polarization state (state 8), which is analogous to the initial state 1 but with oppositely directed polarization vectors.

Importantly, the polarization domain, nucleated at the top of the nanoisland, emerging at the segment 3–4 of the polarization loop is separated by discontinuous polarization jumps from 2 to 3 and from 4 to 7, corresponding to the down-converging and up-diverging polarization states, respectively. Hence, these domains represent thermodynamically metastable states that may be stabilized by specific field-switching pulse sequences. The experimental image ii in panel (a) may correspond to one of these metastable states. The non-equilibrium intermediate dynamic states, such as states 4, 5, and 6, which emerge during the jump from state 4 to state 7, can also be stabilized and pinned by crystal imperfections during the switching process.

The nucleation and subsequent dynamics of the repolarized domain result in the propagation of the domain wall separating up- and down-directed polarization, from the top to the bottom of the nanoisland, as shown in panel (c). The emergence of such a tail-to-tail configuration is usually considered unfavorable and requires large

coercive fields due to the energy-costly depolarization fields produced by the bound charges at the wall. However, as validated by phase-field simulation, the depolarization effects are negligible in our system.

A detailed study of the polarization dynamics presented in panel (d) shows that the downstream and upstream polarization fluxes are not abruptly created at the domain wall boundary but flow from the lateral directions, leading to the continuity of the polarization streamlines without bound-charge formation. This discontinuity-avoiding configuration is a topological state characterized by the singular point V of the vector field of polarization, located at the center of the domain wall[71].

The structure of the vector field in the vicinity of V is shown in panel (e). It is seen that the outgoing vertical polarization flux diverging in the vertical−up and down−directions is compensated by the incoming polarization flux of the spiral-converging branches of the lateral vector field, ensuring the charge neutrality condition $\nabla \cdot \mathbf{P} = 0$ at the singular point. Point V is characterized by the singularity index, calculated as the sign of the Jacobian of the field, $N = ||\partial_i P_j|| = -1$ [72]. The singularity index is an integer number, which is topologically protected, meaning that the point V cannot be deliberately removed by small deformations of the vector field. It is created at the upper surface of the nanoisland and propagates to the bottom surface, topologically characterizing the entire switching process. It is interesting to note that the singular point V is analogous to the Bloch point in magnetic systems, a point-like soliton formation in the magnetization field[73]. However, it possesses an additional constraint of charge neutrality: the flux of polarization through a small sphere surrounding this point must vanish.

To summarize here, the polarization switching of the nanoisland presents a three-dimensional topological evolution of the polarization field $\mathbf{P}$, passing through a series of nonuniform swirled polarization states that can be stabilized by appropriately selected field-modification protocols. Note that with standard measuring conditions (raising/decreasing progressively the voltage and starting from 0 V), the typical hysteresis loop recorded on a nanoisland by PFM switching spectroscopy differs from the calculated one, as shown in Fig. S11.

To conclude, we induced the growth of $BaTiO_3$ trapezoidal funnel-like shaped nanoislands (top size of 30−50 nm) with an excess of Sr in the passivation step of the silicon surface prior to the $SrTiO_3$ epitaxy. STEM analyses show that the $BaTiO_3$ unit cells within an island exhibit a continuous rotation, with opposite rotations with respect to its central axis. The shape of the nanoislands, the strain pattern, and their electrostatic conditions lead to polar center-type down-convergent nanodomains, as revealed by the PFM imaging, PFM data reconstruction, and STEM analyses of the strain and Ti displacements. Phase-field modeling of the polarization pattern of truncated conical-shaped nanostructures confirms the experimental results. Importantly, the phase-field modeling confirms the presence of a whirling component of the polarization around the nanoisland axis, which confers chirality. Considering the analogy between the flow of a liquid and the polarization in ferroelectrics, resulting from their almost divergence-free character, the nanoisland polar texture can be seen as a swirling vortex of polarization flux like a swirling vortex of liquid flows into a narrowing funnel. We show that the topological polar textures can be reversibly switched by the application of an electric field, from center down-convergent to center up-divergent domains. The phase-field simulations provide the three-dimensional dynamical topological evolution of the polarization, passing through a series of nonuniform whirling polarization states.

This work shows that, by appropriately shaping nanostructures, chiral topological textures can be stabilized. Major perspectives of this work are the control of monochiral domains and the demonstration of their manipulation (changing from right-handed to left-handed or vice versa) under an appropriate stimulus. A monochiral state with distinct chirality can be triggered by using an incident beam from a circularly polarized laser. This occurs when cooling the system from an achiral paraelectric state and passing through the ferroelectric transition bifurcation point, leading to the formation of either a left-hand or right-hand chiral polar state[74]. Moreover, the dynamics of the polarization switching under an applied electrical field shows the potential of achieving multiple polar states under appropriate field-modification protocols, which could be leveraged for multi-state analog devices.

## Methods

### Growth of $BaTiO_3$/$SrTiO_3$ on silicon with $BaTiO_3$ nanoislands
$BaTiO_3$/$SrTiO_3$ heterostructures were grown by MBE on p-type Si (001) substrates using a DCA MBE R450 cluster. The Si substrates were treated under UV ($O_3$ generator) at 100 °C for 10 min and then immediately loaded into the load-lock vacuum chamber. The $SrTiO_3$ growth procedure, in situ monitored by reflection high energy electron diffraction (RHEED), involved three main steps[75]. First, the native oxide layer on the Si surface was removed via catalytic desorption of $SiO_x$ by depositing metallic Sr and subsequently annealing under ultrahigh vacuum (UHV) conditions at 870 °C until the 2 × 1 Si surface reconstruction was observed. Additional Sr was then deposited, followed by UHV annealing at 670 °C. At this stage, the regular procedure consists of sending the right amount of Sr to form half a monolayer of Sr, which is monitored by RHEED with the observation of a 2 × 1 Sr surface reconstruction. Here, we introduced an excess of Sr in order to later induce the nucleation of nanoislands at the location where Sr excess aggregated. Then, Ti and Sr were co-deposited from the effusion cells at a substrate temperature of 380 °C under an $O_2$ partial pressure of ~5.0 × $10^{-8}$ Torr. Finally, the films were annealed under UHV at 490 °C to improve the crystallinity. After the epitaxial growth of $SrTiO_3$, the temperature was raised to 650 °C. The 20 nm BTO layer was grown by Ti and Ba co-deposition from the effusion cells under an $O_2$ partial pressure of 5.0 × $10^{-7}$ Torr.

### Structural and microstructural characterization
X-ray diffraction (θ/2θ scans, in-plane scans at χ = 89.5° and phi scans) was performed on the heterostructures using a PANalytical X'Pert PRO diffractometer with a one-dimensional PIXcel detector and Ge (220) monochromator. Cu Kα radiation was used. AFM was performed using a Park Systems NX10 microscope in contact mode with Multi 75Al-G tips. Confocal Raman spectroscopy was done using Horiba LabRam equipment with a grating of 1800 g/mm, a continuous-wave laser of 325 nm wavelength, which was focused through a ×40 objective (NA = 0.49), and an incident laser power of 0.3 mW. The Raman spectrum was taken at room temperature. High-resolution scanning transmission electron microscopy (HR-STEM) was carried out using a probe-corrected JEOL ARM 200 F microscope operated at 200 kV, equipped with a cold field-emission electron gun and a probe aberration corrector allowing a resolution of 0.8 Å at 200 kV in STEM mode. Focused ion beam (Helios Nanolab 600i) was used to prepare cross-sectional lamellae using a standard lift-out procedure. Lamellae were prepared along the [110] crystallographic direction of the Si substrate. Images were recorded using a HAADF detector, which provides imaging of Ba and Ti atomic columns. Stacks of 10 HAADF images were collected with a 21 mrad probe semi-angle, a detector range of 90−370 mrad and a pixel dwell time of 3 μs. The collected HAADF HR-STEM image dataset was processed using the AbStrain method presented in ref. 54 allowing the measurement of strain tensor components with reference to the Bravais lattice of $BaTiO_3$ after correction of scan, sample drift and pixel calibration error distortions. For this, each image presented in a stack was corrected, and then the images were aligned and summed-up, giving rise to an eventual enhanced contrast image free of distortions. We applied the Relative displacement approach, described in detail in ref. 54, for extracting images of Ba and Ti sub-structures and measuring atomic displacements for each sub-

structure with reference to each other, here between the barycenter of the cells constituting the Ba sub-structure and the neighboring atoms of the Ti second sub-structure.

## Polar pattern characterization

PFM was performed using NX10 Park Systems microscope with conductive Pt-coated tips (HQ:NSC18, Mikromash). Dual frequency resonance tracking mode was employed with an external lock-in amplifier (UHFLI, Zurich Instruments). To image the out-of-plane and in-plane polarization components, the vertical oscillation (VPFM) and the lateral torsion (LPFM) of the cantilever were captured with drive frequencies of ~345 kHz and ~600 kHz, respectively, and an ac voltage amplitude of 0.8 V peak to peak.

## PFM data reconstruction

The reconstructions of polarization vector fields from PFM images were done according to the following relations[15]:

$$P_x = A_{l0} \times \cos(\psi_{l0}), \ P_y = A_{l90} \times \cos(\psi_{l90}) \text{ and } P_z = \beta A_V \times \cos(\psi_V)$$

where $A$ is PFM magnitude and $\psi$ is the PFM phase of lateral 0 degree, lateral 90 degree, and vertical scans, respectively. The vertical-to-lateral scaling factor $\beta$ was selected to ensure that the polarization vector is parallel to the sidewalls of the nanoisland.

## Phase-field simulations.

Numerical simulations of polarization texture in ferroelectric nanoislands were based on the minimization of the Ginzburg-Landau-Devonshire free energy functional[76,77] for the pseudocubic ferroelectric material in which the elastic and electrostatic effects are included:

$$F = \int \left( \left[ \alpha_i(T) P_i^2 + \alpha_{ij} P_i^2 P_j^2 + \alpha_{ijk} P_i^2 P_j^2 P_k^2 \right]_{i \leq j \leq k} + \frac{1}{2} G_{ijkl} \left( \partial_i P_j \right) \left( \partial_k P_l \right) \right.$$
$$\left. - \frac{1}{2} \varepsilon_0 \varepsilon_b \left[ (\nabla \varphi)^2 + \delta^{-2} \varphi^2 \right] + (\partial_i \varphi) P_i + \frac{1}{2} C_{ijkl} u_{ij} u_{kl} - C_{ijkl} Q_{klmn} u_{ij} P_m P_n \right) d^3 r. \quad (1)$$

Here we assume the tensor summation over the repetitive indices that takes the cartesian components x, y, z (or 1, 2, 3).

Functional (1) comprises the Ginzburg-Landau energy[78] given in the first square brackets. The second term is the polarization gradient energy[79]. Third and fourth terms represent the electrostatic energy, accounting also the screening effects[80]. The last two terms correspond to the elastic energy. The electrostatic potential and strain tensor are denoted $\varphi$ and $u_{ij}$ respectively. The value of the vacuum permittivity $\varepsilon_0$ is $8.85 \times 10^{-12}$ CV$^{-1}$ m$^{-1}$ and the value of the background dielectric constant $\varepsilon_b$ is 7.35[81]. The numerical values of the Ginzburg-Landau expansion coefficients $\alpha_{ijk}$, gradient energy coefficients $G_{ijkl}$, elastic stiffness tensor $C_{ijkl}$ and tensor of electrostrictive coefficients $Q_{ijkl}$ are given below.

The distribution of the electrostatic potential $\varphi$ and the elastic strains $u_{ij}$ is found from the respective electrostatic (with screening) and elastic equations:

$$\varepsilon_0 \varepsilon_b [\nabla^2 \varphi - \delta^{-2} \varphi] = \partial_i P_j \quad (2)$$

$$C_{ijkl} \partial_i (u_{kl} - Q_{klmn} P_m P_n) = 0 \quad (3)$$

The coefficients of the Ginzburg-Landau expansion for BaTiO$_3$ at room temperature were taken from the sixth-order functional set given in ref .[81] are as follows:

$\alpha_1(T) = -0.02772 \times 10^5$ C$^{-2}$m$^2$N, $\alpha_{11} = 0.1701 \times 10^8$ C$^{-4}$m$^6$N, $\alpha_{12} = -0.3441 \times 10^8$ C$^{-4}$m$^6$N, $\alpha_{111} = 8.004 \times 10^9$ C$^{-6}$m$^{10}$N, $\alpha_{112} = 4.47 \times 10^9$ C$^{-6}$m$^{10}$N, $\alpha_{123} = 4.91 \times 10^9$ C$^{-6}$m$^{10}$N, $Q_{1111} = 0.1104$ C$^{-2}$m$^4$, $Q_{1122} = -0.0452$ C$^{-2}$m$^4$, $Q_{1212} = 0.029$ C$^{-2}$m$^4$, $C_{1111} = 2.75 \times 10^{11}$ m$^{-2}$N, $C_{1122} =$

$1.79 \times 10^{11}$ m$^{-2}$N, and $C_{1212} = 0.543 \times 10^{11}$ m$^{-2}$N. The gradient coefficients $G_{1111} = 0.51 \times 10^{-11}$ C$^{-2}$m$^4$N, $G_{1122} = -0.02 \times 10^{-11}$ C$^{-2}$m$^4$N, and $G_{1212} = 0.02 \times 10^{-11}$ C$^{-2}$m$^4$N, were taken from ref. 81.

The Thomas-Fermi screening length is taken $\delta \approx 1 - 3nm$[82]. Notably, the test simulations conducted using the alternative set for the eighth-order functional, which coefficients are given in ref. 81, yielded qualitatively similar behavior.

The non-linear differential relaxation equation is employed to identify the free energy minima (1):

$$-\gamma \frac{\partial P}{\partial t} = \frac{\delta F}{\delta P} \quad (4)$$

Here $\gamma$ is a time-scale parameter which is taken to be equal unity. The non-linear part of the equations is accompanied by two linear systems of equations defined by the screened Poisson Eq. (2) and the equation of linear elasticity (3).

The phase-field simulations were performed using the FEniCS software package[83]. Three-dimensional trapezoidal computational regions of thickness $d = 20$ nm and with the upper and lower circular basses of diameters $D_u = 50$ nm and $D_l = 35$ nm, respectively, are represented by unstructured tetrahedral finite element meshes. The meshes were created with the 3D mesh generator gmsh[84]. The solutions for $P$, $\varphi$ and $u_{ij}$ were sought in the functional space of piecewise first-order polynomials. The electrostatic potential $\varphi = 0$ and $\varphi = -E_d$ (where $E$ is the applied field) was fixed at the lower and upper surfaces of the nanoisland. The nanoisland was flanked laterally by the dielectric environment with a dielectric constant $\varepsilon_m \approx 100$, simulating the amorphous BaTiO$_3$. Note that simulations with a lower dielectric constant, such as 10, show that the overall behavior remains similar, with no significant changes to the polarization distribution, including the polarization aligning tangentially to the sidewalls of the nanoisland. The compressive strain $u_0 = -0.004$ was fixed at the lower surface. Free boundary conditions were applied for the polarization, strain and electrostatic potential at other surfaces.

The approximation of the time derivative on the left-hand side of Eq. (4) was accomplished by BDF2 variable time stepper[85]. The initial condition for paraelectric polarization state at the first-time step is a random distribution of the individual polarization vector components in the range of $-10^{-6}$ to $10^{-6}$ C m$^{-2}$. Newton method with line search was used to solve the non-linear system arising from Eq. (4). To solve the linear system on each non-linear iteration and systems defined by Eqs. (2) and (3), the generalized minimal residual method with restart was used[86,87].

## Reporting summary

Further information on research design is available in the Nature Portfolio Reporting Summary linked to this article.

## Data availability

The PFM data, 2D reconstruction and modeling data generated in this study have been deposited in a Zenodo repository available at https://zenodo.org/records/14017804 (ref. 88). The STEM data /EELS data that support the findings of this study are available upon reasonable request from nikolay.cherkashin@cemes.fr.

## Code availability

The scripts *AbStrain* and *Relative displacement* applied in the Gatan Microscopy Suite (GMS) application and both used in this study are being implemented and commercialized by HREM Research Inc. company. They will be available at www.hremresearch.com. The source code used for the simulations is available from anna.r-azumnaya@ijs.si upon reasonable request.

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

## Acknowledgements

HZB and CEMES authors acknowledge funding from the Deutsche Forschungsgemeinschaft and Agence Nationale de la Recherche within the project FEAT (DFG 431399790/ANR–19-CE24-0027-01). C.D. and D.J.K. acknowledge funding by the European Union (ERC Advanced Grant, LUCIOLE, project number 101098216). Views and opinions expressed are, however, those of the authors only and do not necessarily reflect those of the European Union or the European Research Council Executive Agency. Neither the European Union nor the granting authority can be held responsible for them. HZB authors acknowledge the cooperation and funding of Park Systems. HZB authors acknowledge the Center for Correlated Microscopy and Spectroscopy (CCMS) at HZB. The research of A.R. was financially supported by the EU HORIZON-WIDERA –2022-TALENTS–FerroChiral, project number 101090285. Y.T. and I.L. acknowledge the H2020-MSCA-RISE-MELON, project number 872631. They also acknowledge the granted access to high-performance computing resources of Plateforme MatriCS within the University of Picardie Jules Verne, co-financed by the European Regional Development Fund (FEDER) and the Hauts-De-France Regional Council.

## Author contributions

C.D., I.O., and D.J.K. initiated the study. C.D. coordinated the study. D.J.K. developed the method for the nanoisland growth and grew the samples by MBE, with the assistance of S.W., and V.H. L.Z. prepared the FIB lamellae. S.S.C. performed the STEM experiments. S.S.C. and N.C. analyzed the STEM images and performed the Abstrain and Relative displacement analyses. I.O. performed the PFM measurements with contributions from D.J.K. I.T., A.R., and I.L. performed the PFM data reconstruction, chirality calculations, and phase-field modeling. C.D., D.J.K., I.O., S.S.C., N.C., I.L., I.T., and A.R. discussed the results. The manuscript was written by I.O., C.D., D.J.K., S.S.C., N.C., and I.L. All co-authors edited the manuscript.

## Funding

## Competing interests

The authors declare no competing interests.
