## [Transparent Peer Review file · Nature Communications]

Switchable topological polar states in epitaxial BaTiO₃ nanoislands on silicon

Corresponding Author: Mr Israel Ibukun Olaniyan

Version 0:

Reviewer comments:

Reviewer #1

(Remarks to the Author)

This manuscript provided a comprehensive study of BTO nanoislands on STO/Si substrates. The authors used XRD, STEM to understand crystallographic structures of the BTO nanoislands, and vector PFM was employed to analyze their topological configurations and switching behavior. The polarization switching dynamics were also further investigated through phase field simulations. The manuscript is in good shape with well presented data. However there are a few issues need to be clarified before publishing.

1. In Figure 1, authors showed two sets of BFO nanoislands with different densities. (1) the AFM images of these nanoislands are not clear enough and the sizes of the nanoisland are not matching between AFM and SEM figures. I would suggest the authors to provide a higher resolution images either by getting better AFM or a zoom-in SEM images to show the shapes and size of the nanoislands; (2) it would be better for the authors to point out the Nanoisland in figure d and f, as there seems to be several different types of nano particles existing in the samples. (3) The authors only mentioned the density difference is attributed to the excess Sr, however, this is important to understand the formation mechanism of the nanoisland and more explanation is necessary. (4) following last question, how the excess Sr also causes the size differences of nanoislands (figure 1d vs figure 1e).

2. In Figure 2, authors give the cross-sectional STEM of nanoislands. (1) Is this sample from figure 1d or figure 1e? (2) why the STEM data doesn't match either the SEM and AFM of both samples? (in respect of distances between nanoisland vs size of nanoislands) (3) the authors explained the dark regions in the square of figure 2b as the growth planarity violation and appearance of tilt walls of edge dislocations. However, there is also bump shape from substrates and STO interface in Figure S2 under each nanoislands. To exclude the impact from substrates, it would be good to give a EDS or EDX mapping of both low and high magnification at the nanoislands regions.

3. Figure 8, the authors demonstrated the bubble domain being created through external electric field. (1) However, this nanoscale domains are not bubble domains with topological configurations when 180 domain walls can be clearly observed. (2) In addition, in the following phase field simulation (page 16, line 427), authors used another term "cylindrical bubble domains", it is necessary to clarify the term-whether it has topological configuration etc. (3) Further how is stability of these cylindrical domains over time?

Reviewer #2

(Remarks to the Author)

In this manuscript, the authors have grown BTO nanoislands on a Si substrate and discovered vortex-like polar topological phases. The discovery of topological phases is widely reported now, and it is unsurprising to find topological phases in nanoisland due to the geometrical confinement (Nature Nanotechnology volume 13, pages947–952 (2018);J. Appl. Phys. 128, 224103 (2020);Nature Communications volume 12, Article number: 1306 (2021)). In general, I think this manuscript lacks novelty for Nature communications. Here are my comments:

(1) There's no TEM polarization mapping for the cross-section view, this is very important to reconfigure the 3D polarization. The real 3D polarization model can't be built with only the PFM images.

(2) Since there is no actual 3D data, the construction of a chirality map (e.g., Fig. 5c) is meaningless.

- (3) Do any property changes associated with the polarization switching, such as conduction or dielectric property?
- (4) In the phase-field model, The strain $u_0 = 0.004$ was fixed at the lower surface, why? Since the bottom is STO, the strain would be compressive at the bottom rather than tensile.
- (5) Since the bottom is STO, and STO is not considered in the phase-field model, mixed boundary condition with open-circuit boundary condition on the bottom would be more realistic, I would suspect the polarization to point inplane close to the lower boundary.
- (6) "The nanoisland was flanked laterally by the dielectric environment with a dielectric constant of 100, simulating the amorphous BaTiO₃", this is also not a good boundary condition, the lateral side of the nanoisland is vacuum/air from the TEM image, which means it will give rise to large lateral depolarization.

Reviewer #3

(Remarks to the Author)

The current work reports the realization of BaTiO₃ (BTO) nanoislands on silicon with switchable center up/down-convergent polarization domains by an electric field. The piezoresponse force microscopy characterization is thorough out and the creation (via bottom-up approaches) and electrically manipulation of chiral whirling polar textures in BTO nanostructures grown on silicon is interesting and holds promise applications in future. So I'd like to recommend it to publication in the current journal, after the authors considering the following points/questions.

The BTO nanoislands is down-convergent polarization domains as grown and is considered as conical-shape nanoislands giving rise to center-down convergent lateral swirling polarization component to prevent from the formation of bound charges on the side walls. The demonstration or discussion is not sufficient. There are defects in the boundary between the nanoisland and the film as evidenced by the STEM in Fig 2, where the bound charges seem highly to be generated (which could be demonstrated or ruled out by conducting atomic force microscopy). I am also curious about the ferroelectric hysteresis loop (corresponding to the simulated PE loop in Fig 8b) or local PFM hysteresis (if the sample is leaky), which may reveal more details about the origin of the down-convergent polarizations. Alternatively, flexoelectric effects (see H. Lu et al, Science 336, 59-61 (2012) or other papers) could be considered due strain gradient induced by the conical-shape nanoislands.

Label confusion in Fig 1b,c: BTO is believed to be tetragonal with a/c domains, but STO seems not. So BTO and STO should not sharing the same crystallography denotes. By the way, c-domain seems to be dominant from the XRD patterns, and what is role that a/c domains playing in the center up/down-convergent polarization domains?

Other minor points as follows. Line 100 on Page 4: "...monolayer of Sr (2x1 Sr surface) aimed", in which the letter "x" in "2x1" should be symbol "x" "2x1" or "2 by 1". Line 446 on Page 17: The writing of "div P=0" is not mathematical right, which should be the right way as the " $\nabla \cdot P$ " in Line 304 on Page 11.

Other suggestions (which could be considered or not) about the figures: Fig 6 with another polarization pattern occasionally observed seems turns away from the main story, which could be moved to the Supporting Information. Fig 7 seems to be redundant, at least Fig 7a,b have been involved Fig 8a, but indeed Fig 8a may need extra images for Fig 7c if one get rid of Fig 7.

Reviewer #4

(Remarks to the Author)

This manuscript reports the successful fabrication of epitaxial BaTiO₃ nanoislands on silicon, which exhibit stable center-down convergent polarization domains and can be reversibly switched to center-up divergent domains by an electric field. This paper is well written and the experimental results look solid. The authors provided a comprehensive analysis of the polar topologies using piezoresponse force microscopy (PFM), scanning transmission electron microscopy (STEM), and phase-field modeling, which firmly evidences the swirling vortex domain structures. I think this manuscript can meet the scope and requirement of Nature Communication and would suggest to accept if after addressing the following comments:

1. As the authors stated, exotic polar patterns have been reported frequently. These patterns are scientifically interesting since they are not similar to typical ferroelectric domain patterns. However, for people new to this area, it is not clearly introduced how these exotic polar patterns can be used in potential applications. I believe some introduction on the potential use will avail boarder readership of this paper.
2. Lines 67-77, the authors stated that fabricating nanostructures is extremely challenging and discuss the difficulty in top-down and bottom-up approaches. It is clear from the statement that the challenges of the top-down approaches are oxygen loss, compositional change, etc., but it is not clear what is the key challenges in the bottom-up approaches. The authors just stated "only a few realization have been reported", but what is the reason?
3. Abstract: "center down-convergent" and "center-down convergent" both appears in the abstract. Please make the terminology consistent.

Version 1:

Reviewer comments:

Reviewer #1

(Remarks to the Author)

The authors well addressed all my comments and concerns in the revised manuscript. I agree it is now suitable for publication in Nature Communications.

Reviewer #2

(Remarks to the Author)

The authors have answered some of the comments, but still some questions regarding the reply:

1. "To the best of our knowledge, this is the first reported realization of topological polar (and moreover chiral) textures in BaTiO₃ nanostructures" Not true, similar work has been published in NC (Jeong, C., Lee, J., Jo, H. et al. Revealing the three-dimensional arrangement of polar topology in nanoparticles. Nat Commun 15, 3887 (2024).)

2. "The lateral side of the nanoisland is not surrounded by vacuum/air, but rather by a region with a high density of dislocations (this was discussed on page 6, now line 151 of the manuscript)", then you should consider the influence of these dislocations on the local stress conditions on your island in the simulation. The stress state should change a lot with nearby dislocations.

Reviewer #3

(Remarks to the Author)

The authors have provided explanations to my comments and made changes of the figures/text in the revised manuscript. So I am satisfied with the revision and I have no further comments.

Reviewer #4

(Remarks to the Author)

The authors properly addressed all my comments. I have no further comments and suggest acceptance of this manuscript.

Version 2:

Reviewer comments:

Reviewer #2

(Remarks to the Author)

Most of my comments have been addressed, I would recommend adding citations for Jeong et al.

We thank the reviewers for their thorough reading of the manuscript, comments and questions, which we address point by point below.

Reviewer #1

Remarks to the Author: This manuscript provided a comprehensive study of BTO nanoislands on STO/Si substrates. The authors used XRD, STEM to understand crystallographic structures of the BTO nanoislands, and vector PFM was employed to analyze their topological configurations and switching behavior. The polarization switching dynamics were also further investigated through phase field simulations. The manuscript is in good shape with well presented data. However there are a few issues need to be clarified before publishing.

Comment 1: In Figure 1, authors showed two sets of BFO nanoislands with different densities. (1) the AFM images of these nanoislands are not clear enough and the sizes of the nanoisland are not matching between AFM and SEM figures. I would suggest the authors to provide a higher resolution images either by getting better AFM or a zoom-in SEM images to show the shapes and size of the nanoislands;

Response: We now provide a higher-resolution AFM image of two BaTiO₃ nanoislands in the Supporting Information file (Figure S2), including line profiles. The shape of the nanoislands observed in Figure S2 is consistent with the one observed in Figure 1f. Due to the finite size of the AFM tip apex the resulting shape of the nanoislands (typically of 30-55 nm lateral size in Fig. 1f) appears rounded.

Unfortunately, obtaining higher-resolution SEM images was not possible due to sample charging/drift during imaging. If we cover the sample with e.g. C to image the nanoislands, their size (and possibly their shape) will be modified.

The AFM and SEM images shown in Figure 1 now have the same field of view and same scale. The lateral sizes observed by SEM and AFM are similar in Figure 1d,f (SEM: 35 - 60 nm, AFM: 30 - 55 nm) and close (in the lower range) to the one measured in STEM (31-34 nm in Fig. 2a).

Revision in the Supporting Information file:

We have added a figure (Figure S2 in the revised file) to provide a high-resolution AFM image of nanoislands.

Revision in the manuscript:

Page 5: Figures 1(d), (e), (f), (g) now all have the same scale bar and field of view.

Page 5, line 134: “A high-resolution AFM image of two nanoislands is provided in Figure S2 of the Supporting Information together with profile lines across them.”

Figure S2: (a) AFM image showing two nanoislands and (b), (c) their line profiles taken along the dashed lines.

(2) it would be better for the authors to point out the Nanoisland in figure d and f, as there seems to be several different types of nano particles existing in the samples.

Response: Thank you for your suggestion. We are only considering the nanoislands that are protruding by about 5 nm in the AFM images (brightest dots in the SEM images). We now point to some of these nanoislands in Figures 1d and 1f using arrows. We can highlight all of them if this is preferred.

Revision in the manuscript

Page 5, line 133: "... whose density can be tuned by varying the Sr excess in the passivation step prior to SrTiO₃ growth (arrows are highlighting some of them in Figure 1d and 1f)."

Figures 1d and 1f and the figure caption have been modified.

Fig. 1: Structural characterization of the sample. **a** Schematic diagram of the epitaxial BaTiO₃/SrTiO₃/Si stack with nanoislands, **b** $\theta/2\theta$ X-ray diffraction pattern of the heterostructure, **c** in-plane X-ray diffraction pattern of the heterostructure, **d, e** SEM and **f, g** AFM images showing the two different densities of BaTiO₃ nanoislands for two different Sr fluxes in the passivation step. Some of the nanoislands in (d) and (f) are highlighted with arrows.

(3) The authors only mentioned the density difference is attributed to the excess Sr, however, this is important to understand the formation mechanism of the nanoisland and more explanation is necessary.

Response: For the growth of atomically smooth epitaxial SrTiO₃ on Si by MBE, the reconstructed (2×1) Si surface is passivated by Sr (to avoid the formation of a SiO₂ amorphous oxide that would preclude any epitaxy on top). The Sr flux is adjusted following the RHEED pattern to form a well-ordered ½ monolayer (½ ML) corresponding to a Sr (2×1) surface. The Sr atoms sit in the trenches formed by the Si dimer rows, in the center of four dimers (see Fig. R1 top view at bottom) [Först *et al.*, Nature 427, 53 (2004)]. For ½ ML coverage, all dangling

bonds of the surface dimers are fully occupied, making this surface robust to oxidation (similarly to the H-terminated Si surface after HF last).

We provide slightly more Sr than needed to form the $\frac{1}{2}$ ML of Sr. It has been shown by DFT calculations [Ashman *et al.*, Phys. Rev. B 69, 075309 (2004) and Potočník *et al.*, Appl. Surf. Sci. 471, 664 (2019)] that Sr will then occupy a position located in between two dimers, on top of a dimer row. This position, or site, named site D in [Ashman 2004, Potočník 2019]) is shown below in Fig. R2 and Fig. R3 (c) and (d). In the study by Potočník *et al.*, the passivation of the Si surface is performed by pulsed laser deposition and adatoms of Sr are evidenced by scanning tunnelling microscopy imaging as shown in Fig. R3(f) (they are non-intentional in this study). When bringing additional Sr compared to the ideal Sr $\frac{1}{2}$ ML so that Sr atoms occupy D sites, the Sr atoms form protrusion on the Si surface, which serve as nucleation centers to form the nanoislands. Our STEM study clearly shows that the nanoisland grows from a “bump” that is already present in the SrTiO₃ layer and that originates from the Si surface (Figure 2b of the manuscript on page 7) as a protrusion in the SiO_x interfacial layer (the Sr aggregates are oxidized once oxygen is sent for the growth of SrTiO₃).

The higher the Sr excess, the greater the density of surface defects, which in turn leads to a higher density of nanoislands.

[Figure redacted]

Fig. R1: Atomic structure of the Sr (2×1) surface on Si (1×2) reconstructed surface. Cross view (top) and top view (bottom). Extracted from Figure 1 of Först *et al.*, Nature 427, 53 (2004)

[Figure redacted]

Fig. R2: Surface structure with 1 ML coverage having a surface reconstruction (2×1) – Extracted from Figure 10 of Ashman *et al.*, Phys. Rev. B 69, 075309 (2004)

[Figure redacted]

Fig. R3: Figure 4 from Potočnik *et al.*, Appl. Surf. Sci. 471, 664 (2019): Simulated STM images of the 1×2 Sr/Si(0 0 1): (a) the ideal structure, (b) with two Sr vacancies, and (c) **with two Sr adatoms**. (d) Schematics of the model with marked site A and D. (e) and (f) Magnification of the area I and II of Fig. 3(b) exhibiting an array of Sr vacancies and Sr adatoms, respectively.

Revision in the Supporting Information file:

We have added a figure (Figure S4 in the revised file) to provide the atomic structure of the Sr (2×1) surface without Sr adatoms (Figure S4a) and with Sr adatoms (Figure S4b), prepared with CrystalMaker®.

(a)

(b)

Figure S4: (a) Top view of the atomic structure of the Sr (2×1) surface corresponding to $\frac{1}{2}$ ML Sr coverage of the (2×1) dimer-row reconstructed Si surface. (b) Top view of the same surface with two Sr adatoms shown in light green occupying sites on top of a dimer row in between two dimers.

Revision in the manuscript:

Page 7, line 190: For a standard epitaxial SrTiO₃ film on Si (001), the reconstructed (2×1) Si surface, which consists of dimer rows, is passivated by ½ monolayer (½ ML) of Sr to avoid the formation of a SiO₂ amorphous oxide when oxygen is introduced in the MBE chamber for the SrTiO₃ growth, which would preclude any epitaxy on top⁴⁸. The Sr flux is adjusted, following the RHEED pattern, to form a well-ordered Sr (2×1) surface. The Sr atoms sit in the trenches formed by the Si dimer rows, in the center of four dimers⁵⁵. This is illustrated in Figure S4a of the Supporting Information. We provide slightly more Sr than needed to form the ½ ML of Sr. It was shown by DFT calculations^{56,57} that Sr atoms then occupy a position located in between two dimers, on top of a dimer row, as shown in Figure S4b. In the study by Potočnik *et al.*⁵⁷, the passivation of the Si surface was performed by pulsed laser deposition and non-intentional adatoms of Sr were evidenced by scanning tunnelling microscopy imaging. Here, we provide additional Sr compared to the ideal Sr ½ ML so that Sr atoms occupy D sites and form protusions on the Sr surface, which serve as nucleation centers to form the nanoislands. Our STEM study (see Figure 2b) clearly shows that the nanoisland grows from a “bump” that is already present in the SrTiO₃ layer and that originates underneath from the Si surface as a protusion in the SiO_x interfacial layer. The elemental profiles across such a “bump” are shown in Figure S5 of the Supporting Information, as obtained from the acquisition of the corresponding EELS edges (O-K, Ti-L, Ba-M, Sr-L and Si-K). Below the SrTiO₃ layer (in the “bump”), an additional Sr peak appears, corresponding to the excess Sr brought initially before the SrTiO₃ layer is grown. The bump is composed of a Ti-Sr-Si-O silicate resulting from the interdiffusion between the SiO₂, SrO (the excess Sr during the passivation phase is oxidized as soon as oxygen is sent) and the upper SrTiO₃ layer (diffusion of Ti).

(4) following last question, how the excess Sr also causes the size differences of nanoislands (figure 1d vs figure 1e).

Response: We have observed that the amount of Sr excess influences strongly the density of nanoislands but does not significantly affect their size. Larger islands are observed and likely result from the merging of two or more nanoislands as seen on the SEM images (e.g. Fig 1e). The adatoms of Sr occupy different sites on the top of the dimer rows and form a larger number of nuclei rather than larger nuclei.

Revision in the manuscript:

Page 8, line 213: The density of the nanoislands can be tuned by varying the Sr flux: the larger the Sr excess, the greater the density of Sr aggregates, which in turn leads to a higher density of nanoislands. Nanoislands may eventually merge as shown in the SEM image of Figure 1e.

Comment 2: In Figure 2, authors give the cross-sectional STEM of nanoislands. (1) Is this sample from figure 1d or figure 1e?

Response: The cross-sectional STEM images provided in Figure 2 correspond to the sample shown in Figure 1d. This has been clarified and included in the manuscript.

Revision in the manuscript

Page 7, line 185 (Figure 2 caption): The STEM images were obtained from the sample with the lower density of nanoislands shown in Figures 1d and 1f.

(2) why the STEM data doesn't match either the SEM and AFM of both samples? (in respect of distances between nanoisland vs size of nanoislands)

Response: The distance between large islands in Figure 1d varies from approximately 60 nm to a few hundred of nanometers, which is consistent with the distances observed in the cross-sectional STEM images. In the example given in Figure 2 (and Figure S3), the distance between the two nanoislands is ~ 120 nm. This is of the order of distances that can be observed between two nanoislands on the AFM images (lower corner of Fig. 1f for example).

Regarding the size of the nanoislands, the STEM images give a size of 31 - 34 nm.

We acknowledge that the low-magnification SEM and AFM images may overestimate the size due to their limited spatial resolution and the AFM tip convolution (and potential charging effect in SEM). However, the STEM data match rather well the SEM and AFM / high resolution AFM data for the lower range size of nanoislands observed.

(3) the authors explained the dark regions in the square of figure 2b as the growth planarity violation and appearance of tilt walls of edge dislocations. However, there is also bump shape from substrates and STO interface in Figure S2 under each nanoislands. To exclude the impact from substrates, it would be good to give a EDS or EDX mapping of both low and high magnification at the nanoislands regions.

Response: The nanoislands originate from Sr aggregates induced by the excess Sr deposited before the initial SrTiO₃ growth (as explained in detail above in Comment 1(3)). As seen in Figures 2b and 2c, the bump corresponds to a quasi-pyramid shape island, which induces the growth planarity violation for subsequent BaTiO₃ deposition. We have performed electron energy loss spectroscopy (EELS) measurements to determine the elemental profiles along such bumps and above them (we have not done EDX since Si and Sr signals are difficult to discriminate). The double Sr peaks confirm that there is an excess Sr brought before the SrTiO₃ layer is grown, located at the "bump". The bump is composed of a Ti-Sr-Si-O silicate resulting from the interdiffusion between the SiO₂, SrO (the excess Sr during the passivation phase is oxidized as soon as oxygen is sent for the subsequent growth of SrTiO₃) and the upper SrTiO₃ layer (diffusion of Ti – Ti easily forms titanium silicides and silicates).

Revision in the manuscript:

Page 8, line 206: "The elemental profiles across such a "bump" are shown in Figure S5 of the Supporting Information, as obtained from the acquisition of the corresponding EELS edges (O-K, Ti-L, Ba-M, Sr-L and Si-K). Below the SrTiO₃ layer (in the "bump"), an additional Sr peak appears, corresponding to the excess Sr brought initially before the SrTiO₃ layer is grown. The bump is composed of a Ti-Sr-Si-O silicate resulting from the interdiffusion between the SiO₂, SrO (the excess Sr during the passivation phase is oxidized as soon as oxygen is sent) and the upper SrTiO₃ layer (diffusion of Ti)."

Revision in the Supporting Information file:

Figure S5 has been added.

Figure S5: (a) HAADF image of a nanoisland with a bump clearly seen underneath and (b) elemental profiles of O, Ti, Ba, Sr and Si obtained from acquisition of the corresponding EELS edges (O-K, Ti-L, Ba-M, Sr-L and Si-K) along the orange arrow in (a).

Comment 3: Figure 8, the authors demonstrated the bubble domain being created through external electric field. (1) However, this nanoscale domains are not bubble domains with topological configurations when 180 domain walls can be clearly observed. (2) In addition, in the following phase field simulation (page 16, line 427), authors used another term “cylindrical bubble domains”, it is necessary to clarify the term-whether it has topological configuration etc.

Response: We are sorry for this terminology confusion. Bubble domains indeed refer to topological configurations (this term has been used in various contexts, including skyrmions, merons, etc.).

Experimentally, (Fig. 8(a) panel ii) of the original manuscript – now Fig. 7), the PFM images of the intermediate state show the presence of both out-of-plane and in-plane polarization components within the domains (not only 180° out-of-plane component). To more accurately reflect this, we have revised the terminology in our manuscript to refer to these domains as “bubble-like domains” rather than bubble domains.

Regarding the phase field simulation discussion, specifically in the sentence “Furthermore, depending ..., these round-shaped nucleations with oppositely oriented polarization may pierce the nanoisland, forming a cylindrical ~~bubble~~ domain at the central part of the nanoisland” mentioned by the reviewer, the type of domain that was discussed was a cylindrical domain with Bloch domain wall (indeed not a bubble domain as, except for the Bloch domain wall, there are no lateral components). This sentence was originally added to refer to another possibility reported in ref. 22 (now ref. 24) but might be confusing since it refers to another scenario and not to our experimental data. Therefore, we have removed the sentence.

Revision in the manuscript:

Figure 8 is now Figure 7 since previous Fig. 6 was moved to the Supplementary Information section

Page 16, lines 411: “bubble-like” was added: “...the V-phase signal turns into a bubble-like core-shell pattern”.

Pages 16, Lines 414, 420, 423, 425: bubble was replaced by “bubble-like”

Page 17, Figure caption 7: bubble was replaced by “bubble-like”

Page 18, line 474: We have removed the sentence: “Furthermore, depending on the system's geometrical and material parameters, these round-shaped nucleations with oppositely oriented polarization may pierce the nanoisland, forming a cylindrical bubble domain at the central part of the nanoisland”.

(3) Further how is stability of these cylindrical domains over time?

Response: They were stable for a test period of 8 hours. We have added this information in the manuscript.

Revision in the manuscript:

Page 16, line 424: is a transitional state between the two types of center-type domains, which is stable in time, as it could be imaged again after eight hours,

Reviewer #2

Remarks to the Author: In this manuscript, the authors have grown BTO nanoislands on a Si substrate and discovered vortex-like polar topological phases. The discovery of topological phases is widely reported now, and it is unsurprising to find topological phases in nanoisland due to the geometrical confinement (Nature Nanotechnology volume 13, pages947–952 (2018);J. Appl. Phys. 128, 224103 (2020);Nature Communications volume 12, Article number: 1306 (2021)). In general, I think this manuscript lacks novelty for Nature communications.

Response: We respectfully disagree with the comment regarding novelty.

Finding topological phases in nanostructures in general has been actually predicted in the early 2000 but was reported experimentally much more recently. To the best of our knowledge, this is the first reported realization of topological polar (and moreover chiral) textures in BaTiO₃ nanostructures, and in addition, on silicon substrate. This is a significant advancement, as the integration of ferroelectric materials in next-generation microelectronic devices requires their integration on silicon.

The studies mentioned in the comment refer to nanostructures grown on **oxide substrates** (LaAlO₃ and SrTiO₃ substrates) and concern **exclusively BiFeO₃ or PbTiO₃ (or PZT)**.

Moreover, the three studies mentioned involve nanostructures with **lateral dimensions in the range of ~ 200 – 400 nm**.

- Nature Nanotechnology 13, 947 (2018): **BiFeO₃** square shape nanoislands, of **lateral size 200 nm** (“... indicates that *the lateral size of the nanoislands is approximately 200 nm and that the BFO nanoislands (about 40 nm thick) are embedded in the BFO matrix (about 20 nm thick)*.”)
- J. Appl. Phys. 128, 224103 (2020): **PZT** (Pb(Zr_{0.7}Ti_{0.3})O₃) nanoislands of lateral size 400 nm (“...the patterned sample from the R-PZT film, which exhibits a well-arranged ordered array of nanoislands with an average diameter of **~ 400 nm**”)
- Nature Communications 12, 1306 (2021): **BiFeO₃** nanodots of **lateral size 400 nm** (“In this work, BFO nanoisland arrays (~40 nm in height and 400 nm in diameter) were directly patterned from high-quality epitaxial BFO films”)

While topological phases in nanostructures are indeed a growing area of research, we report fully novel and original results:

- Our work employs a bottom-up approach to achieve BaTiO₃ nanoislands **on Si** with significantly smaller lateral dimensions of **~30–60 nm** than those reported in the papers cited in the comment (**an order of magnitude lower than the 200-400 nm lateral size** of the mentioned BFO and PZT nanoislands). Note that this method can be extended to grow similar nanostructures of any perovskite grown on a SrTiO₃ template on Si. Nanostructures of BaTiO₃ with these ultrascale dimensions (30-60 nm) have also not been realized by a top-down approach to the best of our knowledge.
- We report for the first time **topological polar textures in BaTiO₃ nanostructures** – moreover with **electrically switchability** demonstrated.
- Moreover, the unique shape of these nanoislands – mimicking a down narrowing funnel – leads to a **chiral** topological state. This realization is also unique and fully novel.

Here are my comments:

Comment 1: There's no TEM polarization mapping for the cross-section view, this is very important to reconfigure the 3D polarization. The real 3D polarization model can't be built with only the PFM images.

Response: There seems to be some misunderstanding of what is presented in Fig. 2(e). This is a continuous map (color map) of the direction of 2D relative displacements of Ti atoms with respect to the barycentre of Ba (or Sr) cells, which is directly related to the polarization. We have chosen this presentation (rather than the most used one showing arrows) to compare with the 2D rigid body rotation map of the Ba (or Sr) cells (Figure 2(d)), allowing us to get a one-to-one relation between these maps.

Regarding 3D polarization, it is not possible to obtain it from STEM for a thin film using a single lamella. If several lamellae are prepared, there is no guarantee that the different directions observed correspond to a same domain.

The reconstructed polarization vector from the PFM data gives a 2D representation of the polarization. We have explicitly written that the P_X, P_Y and P_Z components of the PFM data is an average, at each X,Y location, over the 20 nm thickness of the nanoisland.

The 3D polarization model is built from phase field modelling with input of the PFM data.

Revision of the manuscript:

We have mentioned more explicitly that the reconstructed PFM vector is a 2D representation and that the 3D polarization texture is obtained from phase field modelling with input of the PFM data.

Page 11, line 285: Further insight into the 3D polarization texture in the nanoislands is given by phase field modelling based on the PFM vector reconstruction, as illustrated in Figure 5. The set of analysed vertical, and 0°- and 90°-rotated horizontal, amplitude and phase PFM images is shown in Figure S8 of the Supporting Information. The PFM polarization vector reconstruction (P_x, P_y, and P_z) - for each spatial location X,Y - is a weighted average over the thickness of the nanoisland (with a large contribution from the surface and from underneath the surface).

Comment 2: Since there is no actual 3D data, the construction of a chirality map (e.g., Fig. 5c) is meaningless.

Response: We apologize for any confusion related to the legend in Fig. 5c. The figure actually presents the chirality **indicator** χ' , not the volume chirality. As defined in the text, this indicator is derived from the 2D reconstructed PFM vector, which originates from the depth-averaged 3D polarization distribution across the nanoisland. Therefore, the referee is correct in noting that it cannot be considered a representation of the 3D chirality distribution. We use χ' as an indicator (as it was written in the original manuscript on page 13, now line 321) of the presence of chirality in the volume. Specifically, if χ' is not equal to zero, it suggests that the volume configuration is chiral. However, the inverse is not necessarily true. Positive and negative volume chirality, if present, can cancel each other out by averaging, resulting in $\chi'=0$. In response to the reviewer's remark, we have updated the legend of Fig. 5c accordingly to have the word “indicator” and added a sentence in the manuscript to clarify the meaning of χ' .

Revision in manuscript

Page 12, line 302: c Top view of the reconstructed polarization field. The color map shows the normalized chirality **indicator**, χ' .

Page 13, line 319: Specifically, if χ' is not equal to zero, it suggests that the volume configuration is chiral. However, the inverse is not necessarily true. Positive and negative volume chirality, if present, can cancel each other out by averaging throughout the nanoisland thickness, resulting in $\chi'=0$.

Comment 3: Do any property changes associated with the polarization switching, such as conduction or dielectric property?

Response: Changes in properties associated with polarization switching, such as conduction or dielectric properties, are not within the scope of this study. This study focuses on the demonstration of switchable topological polar states in BaTiO₃ nanostructures on silicon, never reported so far for such a system.

Comment 4: In the phase-field model, The strain $u_0 = 0.004$ was fixed at the lower surface, why? Since the bottom is STO, the strain would be compressive at the bottom rather than tensile.

Response: Thank you very much for drawing our attention to this typo. A compressive strain $u = -0.004$ at the interface BTO/STO was used in the simulations! We corrected it in the Methods section and added “compressive” to make it very explicit for the reader, A tensile strain, if it occurs, would indeed induce an a-oriented polarization vector, but that is not the case.

Revision in manuscript

Page 23, line 637: The **compressive** strain $u_0 = -0.004$ was fixed at the lower surface.

Comment 5: Since the bottom is STO, and STO is not considered in the phase-field model, mixed boundary condition with open-circuit boundary condition on the bottom would be more realistic, I would suspect the polarization to point inplane close to the lower boundary.

Response: We assume that the thin layer of STO exhibits (semi-)conducting properties or electrochemical states at its surface, which effectively screen the bound charges. This assumption aligns with experimental evidence observed in the same BaTiO₃/SrTiO₃/(SiO₂)/Si system grown by molecular beam epitaxy.

- In the paper entitled “Switching of ferroelectric polarization in epitaxial BaTiO₃ films on silicon without a conducting bottom electrode” by C. Dubourdieu *et al.* (Nat. Nano 8, 748 (2013) - doi: 10.1038/NNANO.2013.192) it was shown that 8 and 16 nm films are ferroelectrics. A vertical electrically switchable (up/down) polarization was demonstrated. While SrTiO₃ is not a metallic electrode like La_{0.7}Sr_{0.3}MnO₃ or SrRuO₃, it may have a semiconducting character with a doping level that depends on the oxygen vacancy concentration. We have recently used this property to design a new memristive Ferroelectric Field-Effect Memristor as an electronic synapse on silicon (N. Siannas *et al.*, Adv. Funct. Mater. 2023 - doi: [10.1002/adfm.202311767](https://doi.org/10.1002/adfm.202311767)). Moreover, defects in the SrTiO₃ layer can also contribute to the bound charge screening.

- In the paper entitled “Mixed electrochemical–ferroelectric states in nanoscale ferroelectrics” by S.M. Yang, ..., C. Dubourdieu, S.V. Kalinin (Nat. Phys. 13, 812 (2017) - DOI: 10.1038/NPHYS4103), BaTiO₃ ultrathin films of 1.6, 2.8 and 4.0 nm have been shown to exhibit switchable vertical polar states (for the exact same type of system: BaTiO₃/SrTiO₃/(SiO₂)/Si grown by MBE). This is due to the emergence of a mixed electrochemical–ferroelectric (ferroionic) state.

With these previous results, we can confidently confirm that the bottom SrTiO₃ template should not be modelled with an open-circuit boundary condition.

Additionally, the experimental results obtained by STEM (polarization map in Fig. 2e) supports that the polarization points towards the bottom of the nanoisland (not parallel to it).

Comment 6: "The nanoisland was flanked laterally by the dielectric environment with a dielectric constant of 100, simulating the amorphous BaTiO₃", this is also not a good boundary condition, the lateral side of the nanoisland is vacuum/air from the TEM image, which means it will give rise to large lateral depolarization.

Response: The lateral side of the nanoisland is not surrounded by vacuum/air, but rather by a region with a high density of dislocations (this was discussed on page 6, now line 151 of the manuscript). Therefore, our choice of a dielectric constant of 100 for the simulations is justified as this is a highly disordered region, moreover with a reduced lateral dimension of few nanometers. At this dielectric value of 100, the polarization aligns tangentially to the sidewalls. Importantly, simulations with a lower dielectric constant (see below), such as 10, show that the overall behavior remains consistent, with no significant changes to the polarization distribution and with the polarization aligning tangentially to the sidewalls of the nanoisland. This demonstrates that the choice of a dielectric constant of 100 is appropriate and does not compromise the conclusions of the simulations.

Figure R4: Simulated polarization distribution in a BTO nanoisland with a lateral environment dielectric constant of (a) 10 and (b) 100.

Revision in the manuscript: we have added the following sentence

Page 23, line 634: “Note that simulations with a lower dielectric constant, such as 10, show that the overall behavior remains similar, with no significant changes to the polarization distribution, including the polarization aligning tangentially to the sidewalls of the nanoisland.”

Reviewer #3

Remarks to the Author: The current work reports the realization of BaTiO₃ (BTO) nanoislands on silicon with switchable center up/down-convergent polarization domains by an electric field. The piezoresponse force microscopy characterization is thorough out and the creation (via bottom-up approaches) and electrically manipulation of chiral whirling polar textures in BTO nanostructures grown on silicon is interesting and holds promise applications in future. So I’d like to recommend it to publication in the current journal, after the authors considering the following points/questions.

Comment 1: The BTO nanoislands is down-convergent polarization domains as grown and is considered as conical-shape nanoislands giving rise to center-down convergent lateral swirling polarization component to prevent from the formation of bound charges on the side walls. The demonstration or discussion is not sufficient. There are defects in the boundary between the nanoisland and the film as evidenced by the STEM in Fig 2, where the bound charges seem highly to be generated (which could be demonstrated or ruled out by conducting atomic force microscopy).

Response: There is indeed a large density of dislocations in the region at the interface between the nanoislands and the film. This region does not necessarily give rise to a high density of free

carriers available to screen bound charges (and the dielectric permittivity of this region is much lower than that of an epitaxial BaTiO₃). Based on conductive atomic force microscopy (c-AFM) measurements, we can confirm that the boundaries of the nanoislands are not conductive (see images below in Fig. R5), which suggests that bound charges are unlikely to be present at these boundaries.

Figure R5: (a) AFM image and (b) the corresponding c-AFM image of the same region, showing the topography and local conductivity, respectively. No contrast in current is observed at the nanoisland locations or at their sidewalls. (c) for comparison, we show the current map ($10 \times 10 \mu\text{m}^2$) measured on a (100) ErMnO₃ lamella (prepared by FIB) where enhanced conduction can be seen at domain walls.

Revision in the Supporting Information file:

We have added Figure S7 in the Supporting Information file showing the c-AFM map.

Figure S9: (a) AFM image and (b) the corresponding c-AFM image of the same region, showing the topography and local conductivity, respectively. No contrast in current is observed at the nanoisland locations or at their sidewalls.

Revision in the manuscript:

We have added a comment referring to Figure S9 on the absence of conduction in the boundaries between the nanoislands and the film.

Page 13, line 333: “Conductive AFM allows also to rule out the presence of bound charges on the sidewalls, which would otherwise be associated with screening charges at the boundary between the nanoislands and the film. No enhanced conduction is observed in these boundaries as shown in Figure S9 of the Supporting Information.”

(2) I am also curious about the ferroelectric hysteresis loop (corresponding to the simulated PE loop in Fig 8b) or local PFM hysteresis (if the sample is leaky), which may reveal more details about the origin of the down-convergent polarizations.

Response: Thank you for your suggestion. It is not possible to perform “macroscopic” ferroelectric hysteresis loop on the 30-60 nm nanoislands. We have performed local PFM switching spectroscopy hysteresis loops on the nanoislands. A typical piezo-loop is shown below and has been added to the Supporting Information file with a comment in the manuscript. We observed a slope in the region 1.75 – 2.50 V that could be corresponding to region 3-4 of the hysteresis of Fig. 7(b) (as one figure has been moved to the Supplementary, previous Figure 8 is now Figure 7) but the overall shape does not correspond to the calculated hysteresis. As we indicated in the manuscript, the switching pathway described in Figure 7a was not systematically observed (line 418 “A notable switching pathway in the domain pattern was observed in some cases during the electrical switching from a down-convergent to an up-divergent domain (Figure 7a)”). Moreover, it is obtained when applying a DC bias of -9 V (twice to complete the switching). Hence, we may have to devise a dedicated protocol for the hysteresis loop measurement to be able to reproduce the calculated one of Figure 7b.

Revision in the manuscript

Page 19, line 497: “Note that with standard measuring conditions (raising/decreasing progressively the voltage and starting from 0 V) the typical hysteresis loop recorded on a nanoisland by PFM switching spectroscopy differs from the calculated one as shown in Figure S11.”

Revision in the Supporting Information file:

We have added Figure S11 to show a hysteresis loop measured on a nanoisland.

Figure S11: Local PFM switching spectroscopy performed on a nanoisland. Left: PFM amplitude and phase signals (the phase signal sign here is arbitrary) – Right: Piezoresponse signal.

(3) Alternatively, flexoelectric effects (see H. Lu et al, Science 336, 59-61 (2012) or other papers) could be considered due strain gradient induced by the conical-shape nanoislands.

Response: We agree that there could indeed be some flexoelectric effects in the nanoislands due to the strain gradient arising from their conical shape, as observed in Figures 2d and 2e. However, the fact that the down-convergent polarization domain can be switched to an up-divergent domain under an external electric field, while the conical shape and associated strain gradient remain unchanged, suggests that the flexoelectric effect may not play a significant role in the polarization behavior.

Comment 2: Label confusion in Fig 1b,c : BTO is believed to be tetragonal with a/c domains, but STO seems not. So BTO and STO should not sharing the same crystallography denotes.

Response: Thank you for pointing out this label confusion. Indeed, BTO and STO should not share the same crystallography notation. We have revised the labels in Figures 1b and 1c to reflect the correct crystallographic notations specific to each material.

Revision in manuscript

Page 5, line 125: Fig. 1b,c has been revised.

(2) By the way, c-domain seems to be dominant from the XRD patterns, and what is role that a/c domains playing in the center up/down-convergent polarization domains?

Response: The a/c domains do not play a significant role in the formation of the observed center up/down-convergent polarization domains. These center-type domains are primarily a result of the conical shape of the nanoislands, which influences the polarization configuration independently of the presence of a/c domains in the film matrix.

Comment 3: Other minor points as follows. Line 100 on Page 4: "...monolayer of Sr (2x1 Sr surface) aimed", in which the letter "x" in "2x1" should be symbol "×" "2×1" or "2 by 1". Line 446 on Page 17: The writing of " $\text{div } \vec{P} = 0$ " is not mathematical right, which should be the right way as the " $\nabla \cdot \vec{P}$ " in Line 304 on Page 11.

Response: Thank you for highlighting these points. We have corrected the typographical issue by replacing "2x1" with the appropriate symbol " 2×1 ". Additionally, we have corrected the mathematical notation " $\nabla\cdot P$ ".

Revision in manuscript

Page 4, line 106: 2×1

Page 4, line 108: 2×1

Page 19, line 485: $\nabla\cdot\vec{P}$

Comment 4: Other suggestions (which could be considered or not) about the figures: Fig 6 with another polarization pattern occasionally observed seems turns away from the main story, which could be moved to the Supporting Information. Fig 7 seems to be redundant, at least Fig 7a,b have been involved Fig 8a, but indeed Fig 8a may need extra images for Fig 7c if one get rid of Fig 7.

Response: Thank you for your suggestions. We have moved Figure 6 to the supplementary information, as it shows a polarization pattern that is not central to the main narrative. Regarding Figure 7, we believe it is necessary to keep it in the main text. As you noted, Figure 8a would require additional images if Figure 7 were moved to the supplementary information. To maintain the clarity and completeness of the discussion, we have decided to retain Figure 7 (now Figure 6) in the main manuscript.

Reviewer #4

Remarks to the Author: This manuscript reports the successful fabrication of epitaxial BaTiO_3 nanoislands on silicon, which exhibit stable center-down convergent polarization domains and can be reversibly switched to center-up divergent domains by an electric field. This paper is well written and the experimental results look solid. The authors provided a comprehensive analysis of the polar topologies using piezoresponse force microscopy (PFM), scanning transmission electron microscopy (STEM), and phase-field modeling, which firmly evidences the swirling vortex domain structures. I think this manuscript can meet the scope and requirement of Nature Communication and would suggest to accept if after addressing the following comments:

Comment 1: As the authors stated, exotic polar patterns have been reported frequently. These patterns are scientifically interesting since they are not similar to typical ferroelectric domain patterns. However, for people new to this area, it is not clearly introduced how these exotic polar patterns can be used in potential applications. I believe some introduction on the potential use will avail boarder readership of this paper.

Response: Thank you for your suggestion. We have added an introduction to the potential applications of exotic polar patterns in the introduction section of the manuscript.

Revision in manuscript

Page 2, line 45: "Their nanoscale dimensions (e.g. < 10 nm for skyrmions) make them attractive for ultra high-density data storage ($> \text{Tb/in}^2$), if one can find a way to write / erase

(or “switch”) the polar pattern they contain. Moreover, polar textures and their walls are regions of emerging properties such as negative capacitance^{17,18} or chirality^{19,20}. Negative capacitance is of particular interest to design low-power field effect transistors²¹. Chirality, if it could be manipulated by an external stimulus (electrical or optical) has numerous applications in electronics or photonics.”

Comment 2: Lines 67-77, the authors stated that fabricating nanostructures is extremely challenging and discuss the difficulty in top-down and bottom-up approaches. It is clear from the statement that the challenges of the top-down approaches are oxygen loss, compositional change, etc., but it is not clear what is the key challenges in the bottom-up approaches. The authors just stated “only a few realization have been reported”, but what is the reason?

Response: We have completed this part, pointing out the advantage of the bottom-up approach as compared to the top-down approach.

Revision in manuscript

Page 3, line 75: “The bottom-up approach, while offering the potential for creating smaller nanostructures with fewer defects, faces significant challenges in terms of achieving precise control over the size, shape, and uniformity of the nanostructures³⁵. Additionally, the complexity of the growth processes and the difficulty in ensuring reproducibility have limited the widespread realization of this approach.”

Comment 3: Abstract: “center down-convergent” and “center-down convergent” both appears in the abstract. Please make the terminology consistent.

Response: Thank you for pointing this out. The terminology is now consistent throughout the text (center down-convergent).

Revision in manuscript

Page 1, line 31: center down-convergent

Page 3, line 90: down-convergent

Page 3, line 94: down-convergent

Page 13, line 379: down-convergent

Page 14, line 393: down-convergent

Page 15, line 398: down-convergent

Page 19, line 506: down-convergent

Page 20, line 515: down-convergent

Page 20, line 516: center up-divergent

Reviewer #1

Remarks to the Author: The authors well addressed all my comments and concerns in the revised manuscript. I agree it is now suitable for publication in Nature Communications.

Response: We thank reviewer #1 for the time and effort taken to review our manuscript and for the positive decision.

Reviewer #2

Remarks to the Author: The authors have answered some of the comments, but still some questions regarding the reply:

Comment 1: "To the best of our knowledge, this is the first reported realization of topological polar (and moreover chiral) textures in BaTiO₃ nanostructures" Not true, similar work has been published in NC (Jeong, C., Lee, J., Jo, H. et al. Revealing the three-dimensional arrangement of polar topology in nanoparticles. Nat Commun 15, 3887 (2024).)

Response: We are aware of the work by Jeong *et al.* (Nat Commun 15, 3887, 2024) earlier this year "Revealing the three-dimensional arrangement of polar topology in nanoparticles". This work is on nanoparticles studied experimentally by advanced STEM methods. We were referring to epitaxial nanostructures grown on a substrate (page 3, line 66: "Ferroelectric epitaxial nanostructures such as nanodots, nanodisks, or nanocylinders would be better suited than superlattices for integration into nanoelectronics devices."). Such nanostructures, in addition to being much more manipulable (to perform piezoresponse force microscopy on nanoparticles, one has to embed them in a polymer matrix), can be integrated into devices in the future (no CMOS clean room will be willing to handle nanoparticles, unless, again, is embedded in a composite materials).

The novelty and originality is:

- Our work employs a bottom-up approach to achieve BaTiO₃ nanoislands **on Si** with significantly smaller lateral dimensions of **~30–60 nm** than those reported in the papers cited in the first comments of reviewer 2 (**an order of magnitude lower than the 200-400 nm lateral size** of the mentioned BFO and PZT nanoislands). Note that this method can be extended to grow similar nanostructures of any perovskite grown on a SrTiO₃ template on Si. Nanostructures of BaTiO₃ with these ultrascale dimensions (30-60 nm) have also not been realized by a top-down approach to the best of our knowledge.
- We report **topological polar textures in BaTiO₃ nanostructures on Si** with **electrically switchability** demonstrated. Moreover, the unique shape of these nanoislands – mimicking a down narrowing funnel – leads to a **chiral** topological state on silicon.

Our work represents the first realization of topological polar textures in BaTiO₃ nanostructures integrated on silicon. This is a significant advancement. The integration of ferroelectric materials in next-generation microelectronic devices requires their integration on silicon.

Comment 2: The lateral side of the nanoisland is not surrounded by vacuum/air, but rather by a region with a high density of dislocations (this was discussed on page 6, now line 151 of the manuscript)", then you should consider the influence of these dislocations on the local stress

conditions on your island in the simulation. The stress state should change a lot with nearby dislocations.

Response: A net of dislocations, such as a net of misfit dislocations at any heterostructure interface that ensure plastic stress relaxation or, in our case, a net of dislocations accommodating tilt at a low angle tilt grain boundary, does not have long-range stress and strain fields surrounding it, while individual dislocations do. That is because, at a certain distance, the correspondent fields mutually cancel. The lower the distance between the dislocation lines, the shorter the cancellation range. Hence, at a distance of only a few nanometers, dislocations that are present around the island have very little effect on its deformation.

Reviewer #3

Remarks to the Author: The authors have provided explanations to my comments and made changes of the figures/text in the revised manuscript. So, I am satisfied with the revision and I have no further comments well addressed all my comments and concerns in the revised manuscript. I agree it is now suitable for publication in Nature Communications.

Response: We thank reviewer #3 for the time and effort taken to review our manuscript and for the positive decision.

Reviewer #4

Remarks to the Author: The authors properly addressed all my comments. I have no further comments and suggest acceptance of this manuscript.

Response: We thank reviewer #4 for the time and effort taken to review our manuscript and for the positive decision.

Reviewer #2

Remarks to the Author: Most of my comments have been addressed, I would recommend adding citations for Jeong et al.

Response: We have added citation to Jeong et al. as well as two other references on topology studies of BaTiO₃ nanoparticles.

Revision in the manuscript

Page 3, line 67: “At the other end of the material spectrum, nanoparticles (0D materials) are considered ideal candidates to host topological polar textures. BaTiO₃ nanoparticles embedded in a conducting non-polar polymer, studied by Bragg coherent diffraction imaging, have been shown to exhibit a 3D vortex that can be manipulated under an electric field²⁵. Recently, BaTiO₃ nanoparticles analyzed by atomic electron tomography using scanning transmission electron microscopy revealed a 3D polar topological ordering²⁶. However, manipulating nanoparticles and using other characterization methods such as piezoresponse force microscopy (PFM) is extremely challenging, often requiring embedding the nanoparticles in a matrix to form a composite material^{25,27}.”